# Chromatin endogenous cleavage provides a global view of yeast RNA polymerase II transcription kinetics

Jake VanBelzen[1], Bennet Sakelaris[2], Donna G Brickner[1], Nikita Marcou[1†], Hermann Riecke[2], Niall M Mangan[2], Jason H Brickner[1]*

[1]Department of Molecular Biosciences, Northwestern University, Evanston, United States; [2]Department of Engineering Sciences and Applied Mathematics, Northwestern University, Evanston, United States

*For correspondence:
j-brickner@northwestern.edu

Present address: †Department of Genetic Medicine, Johns Hopkins School of Medicine, Baltimore, United States

Competing interest: The authors declare that no competing interests exist.

## eLife Assessment

This **valuable** study compares ChIP-seq and ChEC-seq2 techniques to investigate RNA polymerase II (RNAPII) binding patterns in yeast, revealing that ChEC-seq2 captures distinct regulatory events associated with active transcription missed by ChIP-seq. The authors use ChEC-seq2 data to build a stochastic model of RNAPII kinetics, providing **convincing** new insights into transcription regulation and the role of the nuclear pore complex. The paper highlights the importance of careful methodological comparisons in understanding RNAPII dynamics.

**Abstract** Chromatin immunoprecipitation (ChIP-seq) is the most common approach to observe global binding of proteins to DNA in vivo. The occupancy of transcription factors (TFs) from ChIP-seq agrees well with an alternative method, chromatin endogenous cleavage (ChEC-seq2). However, ChIP-seq and ChEC-seq2 reveal strikingly different patterns of enrichment of yeast RNA polymerase II (RNAPII). We hypothesized that this reflects distinct populations of RNAPII, some of which are captured by ChIP-seq and some of which are captured by ChEC-seq2. RNAPII association with enhancers and promoters - predicted from biochemical studies - is detected well by ChEC-seq2 but not by ChIP-seq. Enhancer/promoter-bound RNAPII correlates with transcription levels and matches predicted occupancy based on published rates of enhancer recruitment, preinitiation assembly, initiation, elongation, and termination. The occupancy from ChEC-seq2 allowed us to develop a stochastic model for global kinetics of RNAPII transcription which captured both the ChEC-seq2 data and changes upon chemical-genetic perturbations to transcription. Finally, RNAPII ChEC-seq2 and kinetic modeling suggests that a mutation in the Gcn4 transcription factor that blocks interaction with the NPC destabilizes promoter-associated RNAPII without altering its recruitment to the enhancer.

## Introduction

In eukaryotes, differential expression of the genome is achieved primarily through regulated RNA polymerase II (RNAPII) transcription. Since its discovery (*Roeder and Rutter, 1969*), transcription by RNAPII has been the focus of intense study using a variety of methods. From biochemical, structural, and genetic studies, a consensus has emerged for the mechanism of RNAPII transcription (*Figure 1*; *Schier and Taatjes, 2020*). For genes that are dependent on enhancers, sequence-specific transcription factors (ssTFs) bind to enhancers and recruit coactivators like histone acetyltransferases and chromatin remodelers as well as Mediator (*Fishburn et al., 2005*; *Green, 2005*; *Prochasson et al., 2003*;

**Figure 1.** Schematic of RNA polymerase II (RNAPII)-mediated transcription in *Saccharomyces cerevisiae*. Two alternative mechanisms of RNAPII recruitment are shown: (1) direct recruitment to the promoter and (2) recruitment to the upstream activating sequences (UASs) facilitated by sequence-specific transcription factors (ssTFs) and coactivators such as Mediator, followed by transfer to the promoter. After RNAPII associates with the promoter, TFIIH is recruited, leading to phosphorylation of Serine 5 (inset) in the carboxyl terminal domain by the TFIIH-associated kinase Kin28 and initiation. RNAPII elongation through the transcribed region is associated with phosphorylation of Serine 2 in the carboxyl terminal domain. RNAPII pauses during cleavage and polyadenylation before dissociating. Created with BioRender.com.

*Ptashne and Gann, 1997*). Coactivators facilitate the removal of nucleosomes from the promoter, allowing binding of TFIID (TATA binding protein), which recruits additional general transcription factors (GTFs; TFIIA, TFIIB, TFIIF) and ultimately RNAPII (*Figure 1*). Last, TFIIE and TFIIH are recruited to complete the formation of the preinitiation complex (PIC). Through Mediator, ssTFs interact with RNAPII to stabilize the PIC (*Abdella et al., 2021*; *Richter et al., 2022*). TFIIH stimulates initiation by both unwinding the DNA and by phosphorylating the RNAPII carboxyl terminal domain on Serine 5 (*Figure 1*, inset; *Cadena and Dahmus, 1987*; *Komarnitsky et al., 2000*; *Lu et al., 1991*). In metazoans, regulatory factors (negative elongation factor and DRB-sensitive factor [DSIF]) cause RNAPII to pause after initiation, leading to an accumulation of RNAPII downstream of the transcription start site (TSS) (*Adelman and Lis, 2012*; *Core and Adelman, 2019*). The P-TEF-b kinase releases RNAPII from pausing by phosphorylation of these factors and RNAPII on Serine 2, leading to elongation (*Marshall and Price, 1995*). Finally, transcription of a polyadenylation sequence both causes RNAPII to pause, stimulating cleavage and polyadenylation (*Figure 1*; *Nag et al., 2007*; *Orozco et al., 2002*).

To study transcription in vivo, the most common approach has been chromatin immunoprecipitation (ChIP), in which protein-DNA complexes are stabilized through formaldehyde crosslinking and recovered by immunoprecipitation (*Solomon et al., 1988*). Coupled with next-generation sequencing, ChIP-seq has been widely adopted to explore the genome-wide interactions of RNAPII and co-regulators (*Barski et al., 2007*; *Mikkelsen et al., 2007*; *Welboren et al., 2009*). The occupancy of RNAPII over transcribed regions correlates with nascent transcription. Exonuclease foot printing of RNAPII over DNA (ChIP-exo; *Rhee and Pugh, 2012*) or RNA (NET-seq; *Churchman and Weissman, 2011*) and nuclear run-on (PRO-seq; *Kwak et al., 2013*) have provided high resolution of maps of RNAPII binding to the genome. Together, such methods highlight paused and elongating RNAPII and suggest that very little RNAPII is associated with the promoter in the preinitiation state (*Core et al., 2012*).

The dynamics of RNAPII transcription in vivo has also been explored by tracking single molecules of RNAPII (or co-regulators) or individual transcripts. Such experiments offer a different view of transcription. Fluorescence recovery after photobleaching over arrays of inducible reporter genes reveals that a small fraction (~13%) of the RNAPII molecules that assemble at the promoter initiates transcription (*Darzacq et al., 2007*; *Stasevich et al., 2014*). Monitoring the production of single molecules

of mRNA from either such arrays or single genes suggests that RNAPII elongation rate is ~1000–3000 bp/min and that termination is associated with a prolonged pause (50–70 s; *Larson et al., 2011*; *Zenklusen et al., 2008*). Single-molecule tracking of RNAPII and GTFs reveals that ~40% of RNAPII is chromatin-associated and that when initiation is blocked, the dwell time of RNAPII (presumably at the promoter) is ~10 s (*Nguyen et al., 2021*). Because these observations would predict that RNAPII levels at the promoter and terminator (as well as pausing sites) should be higher than those over the transcribed region, they are difficult to reconcile with the RNAPII enrichments observed by ChIP-seq.

Single-molecule tracking of ssTF and RNAPII binding to enhancers and promoters in vitro offers another important perspective. In yeast nuclear extracts, ssTF binding to enhancers (also called upstream activating sequences [UASs]) has been observed. Consistent with the consensus model, ssTFs stimulate RNAPII and PIC recruitment to a neighboring promoter (*Rosen et al., 2020*). Surprisingly, RNAPII and certain PIC components are recruited by ssTFs even in the absence of a promoter (*Baek et al., 2021*). This suggests that RNAPII is recruited to enhancers/UASs by ssTFs, perhaps through interactions with Mediator, which allows efficient promoter loading of PIC components. However, the association of RNAPII and PIC factors with UASs has not been observed by ChIP-seq.

An alternative to ChIP is chromatin endogenous cleavage (ChEC), in which endogenous proteins of interest are tagged with micrococcal nuclease (MNase; *Schmid et al., 2004*). Their association with the genome can be monitored by permeabilizing cells and addition of calcium to activate MNase (*Schmid et al., 2004*). The cleavage events can be identified by next-generation sequencing (ChEC-seq2; *VanBelzen et al., 2024*; *Zentner et al., 2015*). To-date, ChEC-seq2 has only been performed in budding yeast. For ssTFs and nuclear pore proteins, ChEC-seq2 gives results very similar to ChIP-seq or ChIP-exo (*Ge et al., 2024*; *VanBelzen et al., 2024*). Likewise, ChEC-seq2 with coactivators and Mediator resembles ChIP (*Bruzzone et al., 2018*; *Grünberg et al., 2016*; *Saleh et al., 2022*). However, we find that ChEC-seq2 with RNAPII gives a pattern of enrichment that was notably different from that observed using ChIP-seq. Whereas ChIP shows strong enrichment of RNAPII over the transcribed region and little enrichment over the promoter or upstream, ChEC-seq2 showed strong enrichment of RNAPII over the promoter, UAS, and 3'UTR and little signal over the transcribed region. The ChEC-seq2 enrichment of RNAPII over promoters correlated with both nascent transcription (as measured by SLAM-seq; *Herzog et al., 2017*) and ChIP-seq enrichment of RNAPII over coding regions, suggesting that it reflects active RNAPII. RNAPII association with UAS regions was strongest for genes that recruit coactivators and was dependent on ssTFs.

The occupancy of RNAPII over UASs and promoters from ChEC-seq2, combined with published RNAPII dynamics, allowed us to develop a stochastic model for the global kinetics of RNAPII transcription. This model and ChEC-seq2 data offer insight into the effects of genetic perturbations that block transcription globally and suggests that the nuclear pore complex promotes transcription by stabilizing promoter-associated RNAPII. This work suggests that ChEC captures important regulatory events associated with transcription that are missed by ChIP.

## Results

### ChEC-seq2 and ChIP-seq in *S. cerevisiae* yield distinct RNAPII enrichment patterns

To assess ChEC-seq2 with RNAPII, MNase was inserted at the carboxyl terminus of the endogenous genes encoding the RNAPII subunits Rpo21 (also called Rpb1) and Rpb3 (*Zentner et al., 2015*). These yeast strains along with a control strain expressing soluble, nuclear MNase (sMNase) were grown in rich medium, harvested and permeabilized to induce MNase activity. Genomic DNA was prepared and converted into ChEC-seq2 libraries (*VanBelzen et al., 2024*). For comparison, we selected a high-quality RNAPII ChIP-seq dataset from cells grown in rich medium Rpb1 (*Vijjamarri et al., 2023b*; GEO Accession GSE220578) that used the 8WG16 antibody (*Thompson et al., 1989*), which recognizes the carboxyl terminal domain of Rpb1 (*Komarnitsky et al., 2000*). Finally, to confirm that the MNase fusions did not affect RNAPII association with the genome, we also generated ChIP-seq data using this antibody from the yeast strains with and without Rpb1-MNase (*Figure 2—figure supplement 2*). Over transcriptionally active genes like *ILV5*, ChIP-seq gave strong enrichment of Rpb1 over the transcribed region and terminator and low enrichment over the enhancer/UAS and the promoter (*Figure 2A*, first row). In contrast, ChEC-seq2 with either Rpb1 or Rpb3 showed strong enrichment

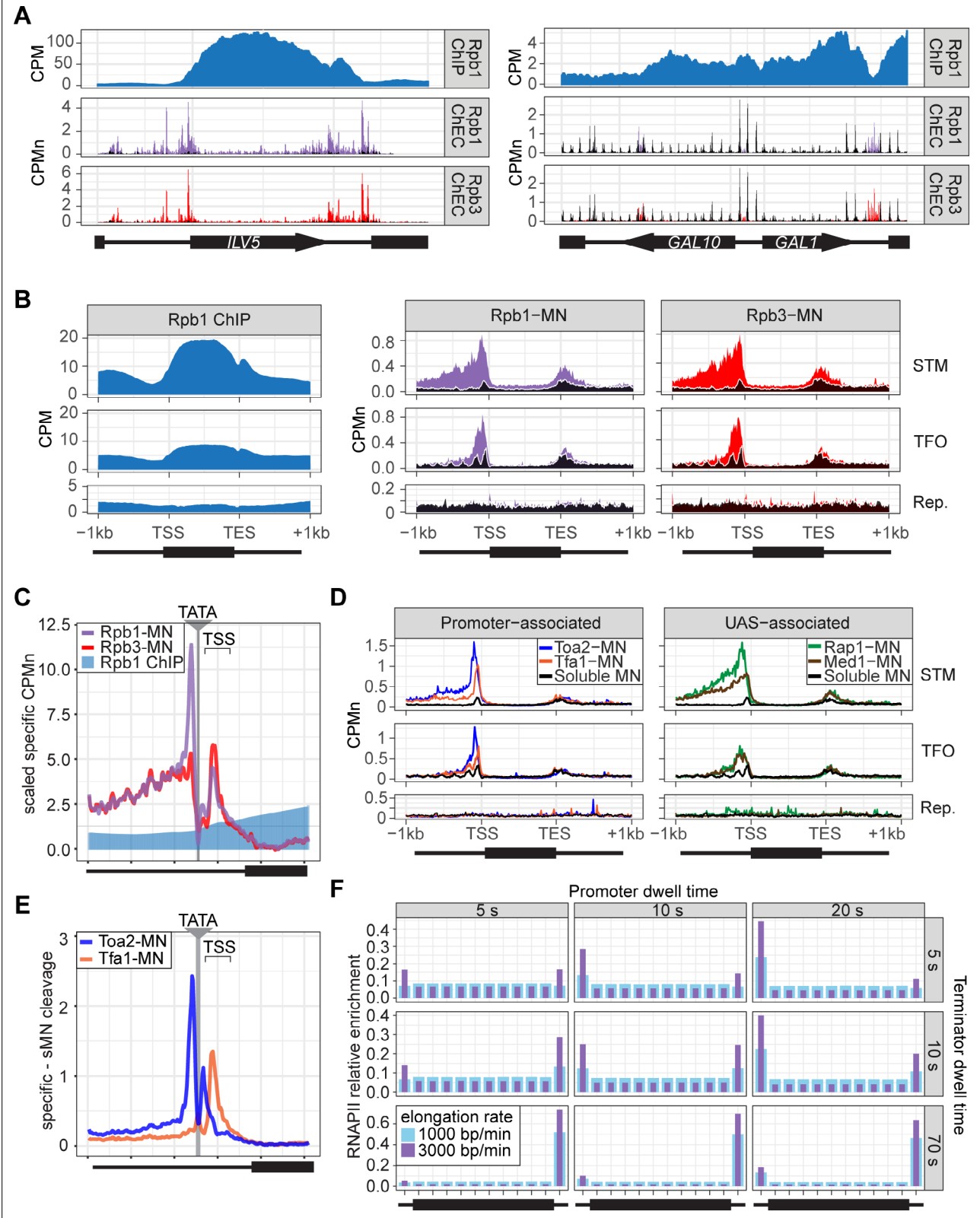

**Figure 2.** ChEC-seq2 and ChIP-seq reveal distinct RNA polymerase II (RNAPII) interactions with the genome. (**A**) Gene plots displaying mean counts per million reads (CPM; *Vijjamarri et al., 2023a*) for ChIP-seq for Rpb1; or CPM-normalized cleavage frequency (CPMn) for ChEC-seq2 for Rpb1-MN or Rpb3-MN over the *ILV5* and *GAL1-10* loci±1 kb. Plots are smoothed using a window of 10 bp and a step size of 5 bp. Signal from soluble micrococcal nuclease (sMNase) is shown in black. (**B**) Metagene plots showing average signal over subsets of genes with distinct expression levels and mechanisms

*Figure 2 continued on next page*

*Figure 2 continued*

of regulation. The average signal from 150 genes with highest expression from STM and TFO classes (*Rossi et al., 2021*) and 84 repressed genes is plotted, along with sMNase (black; genes listed in *Supplementary file 1*). A length-normalized transcript (rectangle), 1 kb upstream of the transcription start site (TSS), and 1 kb downstream of the transcription end site (TES) is shown. Rpb1 ChIP-seq (left), ChEC-seq2 with Rpb1-MN (middle), or Rpb3-MN (right). (**C**) ChIP (Rpb1) and ChEC (Rpb1-MN and Rpb3-MN) signal over 597 TATA boxes from expressed genes ±250 bp (*Supplementary file 1*; *Rhee and Pugh, 2012*). The location of the TATA sequence is indicated with a gray bar and the TSS is 50 bp±39 bp to the right of the center of the TATA. For ChEC data, sMNase was subtracted from the respective specific CPMn. (**D**) Metagene plots as in (**B**) from ChEC-seq2 with Toa2-MN (TFIIA) and Tfa1-MN (TFIIE; left) or the sequence-specific transcription factor (ssTF) Rap1-MN and Med1-MN (Mediator; right) from 287 STM genes containing Rap1-peaks, top 150 expressed TFO genes, or 84 repressed genes. (**E**) Mean CPMn at TATA genes as in (**C**) from ChEC-seq2 with Toa2-MN (blue) or Tfa1-MN (orange). (**A–E**) The averages from three biological replicates. (**F**) Predicted occupancy of RNAPII based on a range of promoter dwell times (5–20 s), elongation rates (1000–3000 bp/min), and termination times (5–70 s). The transcribed region is 1200 bp divided into 10×120 bp bins, flanked by an upstream promoter bin and downstream terminator bin. RNAPII occupancy was simulated using a minimal stochastic model. RNAPII was assumed to be immediately present at the promoter and progressed to the transcript region with a rate inverse to the promoter dwell time. It then progressed along a 1200 bp coding region with the indicated elongation rate and terminated transcription with a rate inverse to the terminator dwell time.

The online version of this article includes the following figure supplement(s) for figure 2:

**Figure supplement 1.** RNA polymerase II (RNAPII) ChEC vs. ChIP.

**Figure supplement 2.** ChIP-seq against Rpb1 vs Rpb1-MN.

at the UAS, promoter, and terminator of *ILV5* and a low enrichment over the transcribed region (*Figure 2A*, second and third rows; compare with sMNase in black). However, over the repressed *GAL1-10* locus, both ChIP-seq and ChEC-seq2 show background enrichment for RNAPII (*Figure 2A*, right). Notably, sMNase cleavage over *GAL1-10* reflects both unprotected linkers between well-positioned nucleosomes and nucleosome depletion upstream of promoters (*Chereji et al., 2019*; *Lee et al., 2004*; *Figure 2A*, right). This pattern was unrelated to the trimming of mapped reads to the first base pair (compare with untrimmed tracks in *Figure 2—figure supplement 1A*), the normalization of transcript length used in metagene plots (enrichment over promoters and the 5' end of genes in *Figure 2—figure supplement 1B*; see Methods), or the presence of the MNase fusion (*Figure 2—figure supplement 2*). Globally, while both ChIP-seq and ChEC-seq2 showed positive Spearman correlation with nascent transcription (measured by SLAM-seq), different regions of genes correlated best with nascent mRNA (*Figure 2—figure supplement 1C*). Nascent transcription correlated best with the enrichment of RNAPII over the promoter from ChEC-seq2 and the enrichment of RNAPII over the transcribed region and terminator from ChIP-seq. Thus, both ChIP-seq and ChEC-seq2 with RNAPII show enrichments that correlate with transcriptional activity, but these two methods reveal complementary interaction patterns.

Different classes of RNAPII-transcribed yeast genes show distinct mechanisms of transcriptional regulation (*Rossi et al., 2021*). To more precisely define the differences between ChIP and ChEC, we compared ChIP-seq with ChEC-seq2 over three such classes: (1) genes that bind ssTFs and coactivators such as S̲AGA, T̲up1, M̲ediator, SWI/SNF (STM), (2) genes bound to ssTFs but not coactivators (t̲ranscription f̲actors o̲nly, TFO), and (3) a set of 84 genes that showed no detectable nascent transcription, based on SLAM-seq (repressed). Because these different classes of genes are expressed at different levels (*Figure 2—figure supplement 1D*), we focused on the most highly expressed 150 genes from the STM and TFO classes. Metagene plots of mean RNAPII ChIP-seq over each of these sets of genes reveal strong enrichment over the transcribed region for the STM genes and, to some extent, for the TFO genes, with a notable dip over the promoter (*Figure 2B*, left). Metagene plots of RNAPII ChEC-seq2 showed a strong enrichment over the promoter for both STM and TFO genes and over the UAS for STM genes (*Figure 2B*, middle and right). RNAPII was not enriched over repressed genes by either method.

To better understand the ChEC patterns upstream of TSSs, mean cleavage by RNAPII was

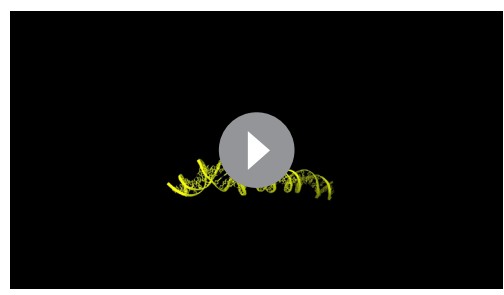

**Video 1.** Animation of the accessibility of DNA to cleavage during assembly of the closed preinitiation complex, based on PDB 7nvs (*Aibara et al., 2021*). https://elifesciences.org/articles/100764/figures#video1

plotted at higher resolution by aligning to 597 high-confidence TATA boxes upstream of expressed genes (based on SLAM-seq), oriented so that the TSS is 50 bp ± 39 bp to the right (*Figure 2C*; ±250 bp). Because sMNase cleaves the TATA boxes strongly (*Figure 2—figure supplement 1E*) - reflecting either increased accessibility or the T/A sequence preference of sMNase (*Dingwall et al., 1981*; *Hörz and Altenburger, 1981*) - we subtracted the sMNase cleavage from specific cleavage frequency (*Figure 2C*). Both Rpb1-MN and Rpb3-MN produced cleavage peaks ~17 bp upstream and ~34 bp downstream of the TATA box, although their relative intensities were different (*Figure 2D*). In contrast, Rpb1 ChIP-seq signal was low over the TATA and TSS (*Figure 2C*).

The ChEC-seq2 signal for RNAPII over the UAS region correlates with recruitment of coactivators upstream of STM genes, but not upstream of TFO genes (*Figure 2B*, middle and right), arguing that it is not an artifact of nearby promoters or genes. To better understand the ChEC-seq2 signal over promoters and UAS regions, we mapped proteins expected to interact with the promoter (PIC components TFIIA [Toa2] and TFIIE [Tfa1]) or the UAS (the Rap1 ssTF and Mediator). For this comparison, we selected 287 STM genes near high-confidence Rap1 sites (*VanBelzen et al., 2024*). While the PIC interacted strongly with the promoter region of both STM and TFO genes, Rap1 and Mediator interacted strongly with the UAS region of STM genes (*Figure 2D*). Rap1 and Mediator also showed a low level of enrichment upstream of the promoter region of TFO genes (*Figure 2D*). Thus, ChEC-seq2 showed promoter enrichment of PIC components and UAS enrichment of TFs and Mediator.

When mapped over TATA sites, TFIIA (Toa2-MN) produced a major cleavage peak ~12 bp upstream and a minor peak ~12 bp downstream from the TATA box (*Figure 2E*). TFIIE (Tfa1-MN) showed the strongest peak ~34 bp downstream of the TATA (*Figure 2E*). These data suggest that ChEC-seq2 reflects the arrangement of TFIIA, RNAPII, and TFIIE within the PIC: TFIIA interacts with DNA immediately upstream of TBP, RNAPII binds on both sides of TBP, and TFIIE binds downstream of TBP (see *Video 1*; *Aibara et al., 2021*; *He et al., 2013*; *Schilbach et al., 2021*). Also, consistent with an ordered assembly of the PIC, the peak of TFIIA cleavage 12 bp downstream of the TATA box is absent in the RNAPII and TFIIE ChEC data, suggesting that TFIIA binds before RNAPII and TFIIE during PIC assembly and that this site becomes protected when RNAPII and TFIIE join (*Video 1*). Together, these data suggest that ChEC-seq2 captures both UAS-associated RNAPII and the PIC.

Given the dramatic difference between ChEC-seq2 and ChIP-seq, we next asked if either pattern is consistent with the dynamics of transcription as described in the literature. Because *S. cerevisiae* lacks promoter-proximal pausing (*Booth et al., 2016*) and has few intron-containing genes that require splicing (*Stajich et al., 2007*), these slow elongation steps are expected to be absent. Therefore, RNAPII initiation and pausing during termination (*Hyman and Moore, 1993*) would represent relatively slow steps compared with the rate of elongation. Both in vivo and in vitro studies in yeast suggest promoter dwell times in the range of approximately 5–20 s (*Baek et al., 2021*; *Nguyen et al., 2021*) a termination time of up to 70 s and an elongation rate between 1000 and 3000 bp/min (*Larson et al., 2011*; *Zenklusen et al., 2008*). Using these ranges, we calculated the predicted RNAPII occupancy over the promoter, the transcribed region, and the terminator for the typical transcribed yeast gene (see Methods; median size of transcribed region = 1.2 kb; *Pelechano et al., 2013*). Of 24 combinations of dwell times and elongation rates tested, 21 predicted higher occupancy at the promoter than over the transcribed region and 21 predicted higher occupancy at terminators than over the transcribed region (*Figure 2F*). While some combinations predicted a relatively flat distribution across the gene with lower levels in the promoter, none of the 24 predicted the strong signal over the transcribed region with promoter depletion characteristic of ChIP-seq. Only very short promoter dwell times (i.e. <1 s) produced the low promoter occupancy seen in ChIP-seq (*Figure 2—figure supplement 1F*). This suggests that ChIP-seq is unable to detect functionally important RNAPII interactions at the promoter and UAS that are detected by ChEC-seq2.

## ChEC-seq2 detects elongating and phosphorylated RNAPII

Next, we performed ChEC-seq2 with the kinases involved in initiation and elongation, as well as the elongation factor Spt5 (part of DSIF). Phosphorylation of the carboxy terminal domain (CTD) of RNAPII regulates its activity and the association of factors involved in splicing, histone modification, and RNA processing. Initiation correlates with phosphorylation of Ser5 of the CTD by Kin28 (Cdk7/ TFIIH kinase; *Komarnitsky et al., 2000*). Elongation is coupled with phosphorylation of Ser2 by Ctk1

(P-TEF-b; CTDK-I; Cdk9; *Cho et al., 2001*) and Bur1 (P-TEFb; *Qiu et al., 2009*), and the association of Spt4/5 (DSIF; *Hartzog et al., 1998*).

Kin28-MN, Ctk1-MN, and Spt5-MN showed strong cleavage over active genes and little enrichment over inactive genes (*Figure 3A*). All three proteins showed maximum cleavage over the promoters of active genes. Kin28 showed significant enrichment over the UAS region of STM genes that was absent from TFO genes (*Figure 3A*, left). The elongation factor Spt5 showed enrichment over both as well as the transcribed region (*Figure 3A*, right). In contrast, Ctk1-MN cleavage was primarily localized to promoters (*Figure 3A*, middle). Higher resolution mapping aligned to TATA boxes confirmed that, while Rpb1 shows peaks of cleavage upstream and downstream of TATA, Kin28, Ctk1, and Spt5 show a single peak downstream, near the TSS (*Figure 3B*). Furthermore, the signal upstream of the TATA was greatest for Kin28, followed by Ctk1 and then Spt5 (*Figure 3B*). This suggests that, while Rpb1 shows interactions at the TSS and upstream, factors involved in initiation and elongation are more enriched with the TSS and over the transcribed region.

To confirm that the ChEC cleavage pattern by Kin28 and Ctk1 reflects their activity, we developed a method to measure RNAPII phosphorylation by ChEC-seq2. Two single chain IgG fragments that recognize phosphorylated Ser2 (Ser2p) RNAPII CTD or phosphorylated Ser5 (Ser5p) RNAPII CTD (Mintbodies) have been expressed as GFP- and SNAP-tagged fusions and shown to localize at transcriptionally active loci in mammalian cells (*Ohishi et al., 2022*; *Uchino et al., 2022*). We constructed Mintbody-MNase (Mb-MN) fusions to detect these phosphorylated forms of RNAPII (*Figure 3C*; α-Ser2p-MN and α-Ser5p-MN). Because binding phosphorylated CTD could compete for critical interactions with RNAPII, we tested several promoters to identify an expression level that produced the smallest growth defect (not shown). Strains expressing the Mb-MNs from the *ADH1* promoter had a minimal growth defect (*Figure 3D*) and cleaved chromatin upon permeabilizing cells and addition of calcium (*Figure 3—figure supplement 1A*). Both α-Ser5p-MN and α-Ser2p-MN give patterns very similar to those produced by their respective kinases; Ser5p was more enriched over promoters and UAS regions, while Ser2p was more evident in the transcribed region (*Figure 3E and F*). To better compare these patterns, we normalized mean cleavage by the mintbodies over promoters, UAS regions, transcribed regions, and 3'UTR regions by each Mb-MN (or sMNase) to cleavage by Rpb1 (*Figure 3G*). Ser5p and Ser2p levels were lower than Rpb1 over the UAS and promoter, but higher than Rpb1 over the transcript and 3'UTR (*Figure 3G*). Furthermore, consistent with their patterns observed by ChIP, the levels of Ser2p were lower than those of Ser5p over the UAS and promoter and higher than those of Ser5p over the transcript (*Figure 3G*). Thus, ChEC-seq2 can reveal RNAPII recruitment, initiation, and elongation during transcription.

## Global transcriptional changes are detected by ChEC-seq2

To further validate the biological significance of RNAPII ChEC-seq2, we examined the effects of an environmental perturbation that results in a large-scale transcriptional change. Cells exposed to 10% ethanol in growth medium show widespread changes in transcription, downregulating hundreds of genes enriched for those involved in ribosome biogenesis (GO: 0042254; blue in *Figure 4A*) and upregulating genes enriched for chaperones (GO: 0009266; red in *Figure 4A*). ChEC-seq2 using Rpb1-MN, Kin28-MN, Ctk1-MN, α-Ser5p-MN, and α-Ser2p-MN captures these changes. These proteins showed increased enrichment over the *HSP104* chaperone gene following ethanol treatment (*Figure 4B*). Likewise, metagene plots over the top 100 induced genes showed increased cleavage by Rpb1-MN, Kin28-MN, Ctk1-MN, as well as their products Ser5p and Ser2p upon ethanol treatment (*Figure 4C and D*, left). Notably, over the transcribed region, enrichment was higher at the 3' end, especially for Ser2p and Ctk1-MN (*Figure 4C and D*, left). In contrast, metagene plots of the average change in cleavage over the 137 ribosomal protein genes showed strong decreases in cleavage by all of these proteins (*Figure 4C and D*, right). The changes in sMNase cleavage were generally the opposite of what we observed with the specific proteins (*Figure 4C and D*, black trace/column). Thus, ChEC-seq2 can capture biologically relevant changes in RNAPII association, its regulators, and its phosphorylation states that reflect large-scale changes in global transcription.

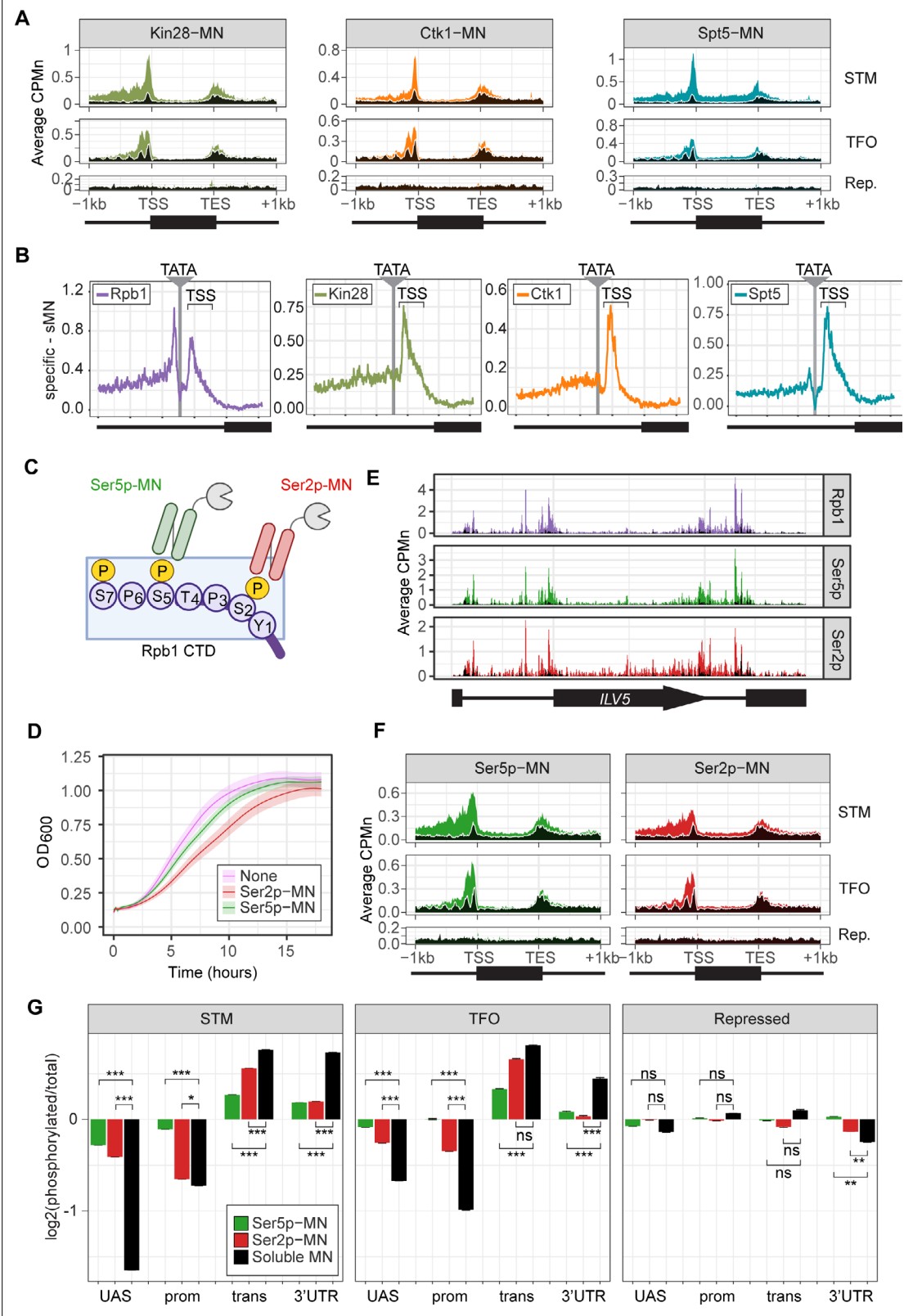

**Figure 3.** ChEC-seq2 to monitor initiation and elongation. (**A**) Metagene plots showing average CPMn for ChEC with the indicated proteins over subsets of genes with distinct expression levels and mechanisms of regulation. The average signal from 150 genes with highest expression from STM and TFO classes (***Rossi et al., 2021***) and 84 repressed genes are plotted (***Supplementary file 1***). Signal from the soluble microccocal nuclease (sMNase) control is shown in black with a white line for contrast. (**B**) Average CPMn for the indicated proteins over 597 TATA boxes ± 250 bp from

*Figure 3 continued on next page*

*Figure 3 continued*

expressed genes (*Supplementary file 1*; *Rhee and Pugh, 2012*). The location of the TATA sequence is indicated with a gray bar and the transcription start site (TSS) is 50 bp ± 39 bp to the right of the TATA. The signal for sMNase was subtracted from each. (**C**) Schematic for Mintbody-MNase constructs. Two single chain variable fragments of IgG specific to phosphorylation of Serine 5 (Ser5p) or Serine 2 (Ser2p) of the carboxy terminal domain (CTD) of RNA polymerase II (RNAPII) (*Ohishi et al., 2022*; *Uchino et al., 2022*) were tagged at their C-termini with MNase. (**D**) $OD_{600}$ of the parent strain (pink), strain expressing Ser5p-MN (green), and strain expressing Ser2p-MN (red) over time in culture (average ± standard deviation). (**E**) Average CPMn from ChEC-seq2 with Rpb1-MN (purple), α-Ser5P-MN (green), and α-Ser2P-MN (red) at *ILV5* ± 1 kb. Plots were smoothed with a step size of 5 and window of 10. (**F**) Metagene plots as in (**A**), but with signal from Ser5p-MN (green; left) and Ser2p-MN (red; right). (**G**) For each protein, the relative enrichment at upstream activating sequence (UAS), promoter, transcript, and 3'UTR regions was calculated and normalized by region length for each gene. The resulting values were normalized to those values for Rpb1-MN and the average from all genes in each group is plotted. Error bars represent the estimated variance between biological replicates from standard deviation (n=3). Differences between ratios and estimated variance were used to calculate a z score and p-value; *p<0.05, **p<0.01, ***p<0.001. All panels represent the average from three biological replicates. Panel C was created with BioRender.com.

The online version of this article includes the following source data and figure supplement(s) for figure 3:

**Figure supplement 1.** Mintbody-directed ChEC.

**Figure supplement 1—source data 1.** TIFF file containing original immunoblots from *Figure 3—figure supplement 1B*.

**Figure supplement 1—source data 2.** Original files for immunoblot from *Figure 3—figure supplement 1B*.

## RNAPII ChEC-seq2 upon chemical-genetic perturbations of transcription

Next, we tested the effect of blocking either PIC formation or initiation on RNAPII/PIC occupancy by ChEC-seq2. PIC formation was blocked by depleting TFIIB using auxin-induced degradation (Sua7-AID; *Figure 5A*) and initiation was inhibited by treating an analog-sensitive allele of Kin28 with the ATP analog CMK (*kin28-is*; *Rodríguez-Molina et al., 2016*). These treatments resulted in strong downregulation of nascent transcription (SLAM-seq; *Figure 5B*) and inhibition of growth (*Figure 5—figure supplement 1*), respectively. ChEC-seq2 with Rpb1 following 20 min of depletion of TFIIB showed a clear decrease of Rpb1-MN cleavage over the promoters of the 150 most highly transcribed STM and TFO genes (*Figure 5C*). TFIIB depletion caused a shift in sMNase cleavage from the TSS downstream (*Figure 5C*). Neither Rpb1-MN nor sMNase cleavage over repressed genes was altered by TFIIB depletion (*Figure 5C*). This suggested that Rpb1 occupancy over the promoters of STM and TFO genes requires TFIIB.

Higher resolution mapping of RNAPII (Rpb1-MN), TFIIA (Toa2-MN), and TFIIE (Tfa1-MN) cleavage over TATA boxes revealed that, upon TFIIB depletion, TFIIA occupancy shifted from the major upstream peak to the downstream peak (*Figure 5D*). RNAPII and TFIIE peaks near the TATA and TSS were lost (*Figure 5D*). This supports the notion that the downstream peak of TFIIA is blocked by RNAPII/PIC binding. Furthermore, the cleavage by Rpb1 upstream of the TATA box was unaffected by depletion of TFIIB (*Figure 5D*, middle), suggesting that TFIIB is required for proper PIC formation over the promoter, but is not required for association with upstream UAS elements.

To test this hypothesis, we mapped RNAPII (Rpb1-MN) cleavage over 896 high-confidence sites for the ssTF Rap1 (*VanBelzen et al., 2024*). Rap1 regulates hundreds of highly expressed genes and RNAPII ChEC-seq2 showed strong enrichment flanking Rap1 sites, while sMNase did not (*Figure 5E*). This correlates with Mediator occupancy (*Figure 5F*). Depletion of TFIIB had no significant effect on RNAPII occupancy over Rap1 sites (*Figure 5E*). Thus, RNAPII recruitment to the promoter is dependent on TFIIB, while RNAPII recruitment to the UAS is not.

Inhibition of *kin28-is* with CMK also lead to a strong decrease of RNAPII over the promoter, transcribed region, and 3'UTR, especially for the STM genes (*Figure 5G*). As expected, this was associated with a strong decrease of Ser5 phosphorylation and Ser2 phosphorylation (*Figure 5G*). Cleavage by α-Ser5p-MN was most strongly decreased at the promoter, while cleavage by α-Ser2p-MN was most strongly decreased at the 3' end of the transcribed region. No changes in cleavage were observed at repressed genes. RNAPII cleavage over 597 TATA boxes near expressed genes was also decreased upon Kin28 inhibition, but this effect was not as strong as that observed upon depletion of TFIIB (*Figure 5H*). Thus, inhibition of Kin28 led to an apparent decrease in total RNAPII and its Ser2 and Ser5 phosphorylated forms from highly expressed genes.

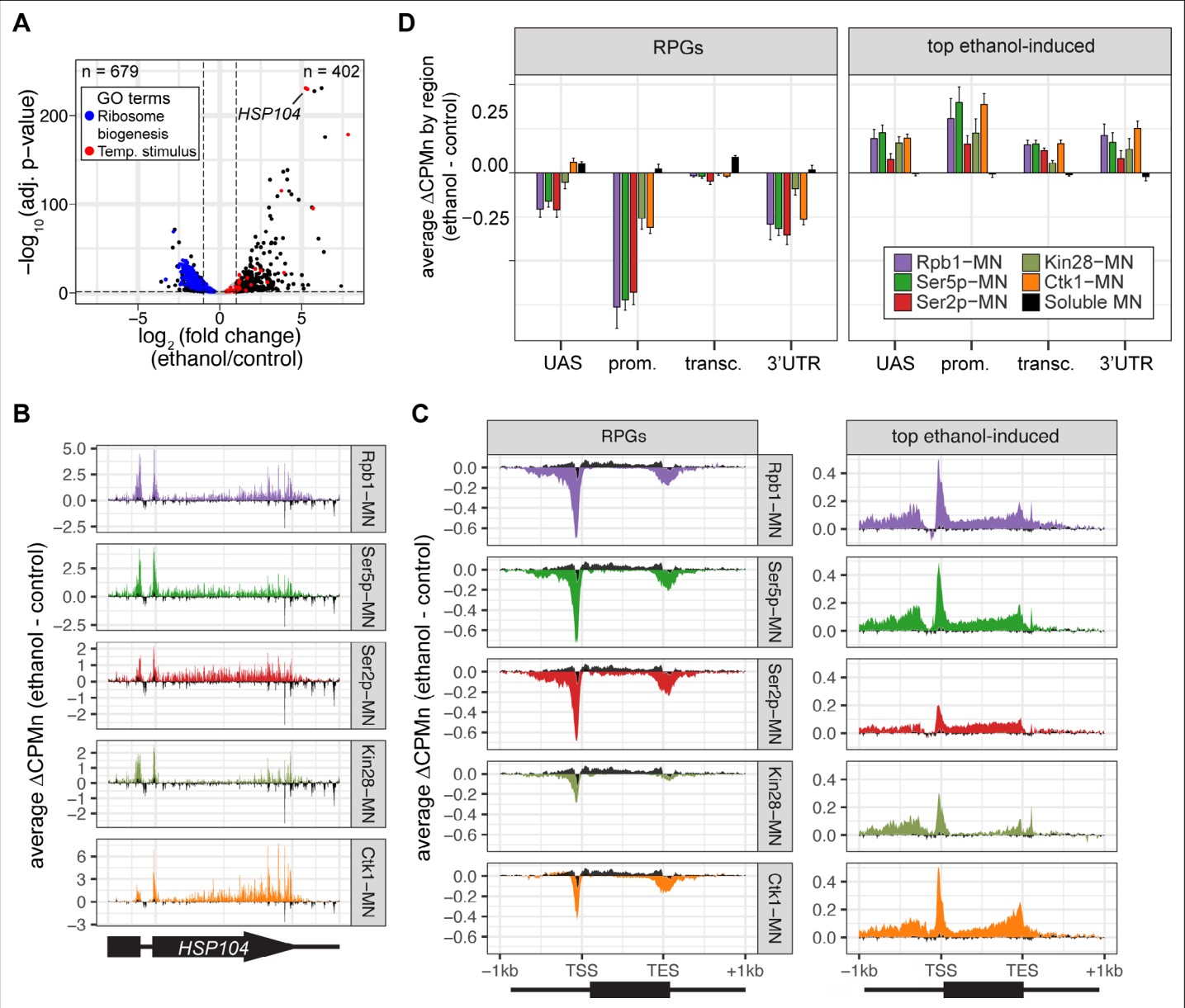

**Figure 4.** Transcriptional response to ethanol stress results in widespread changes in RNA polymerase II ChEC-seq2. (**A**) Volcano plot of fold-change vs. -log$_{10}$ of adjusted p-values of 5295 mRNAs comparing cells treated with 10% ethanol for 1 hr vs. untreated cells. The mRNAs belonging to the most statistically significant terms from Gene Ontology Enrichment analysis of the 402 upregulated mRNA (response to temperature stimulus, GO: 0009266; red) or 679 downregulated mRNA (ribosome biogenesis, GO: 0042254; blue) are highlighted. (**B**) Average change in CPMn over *HSP104* ± 1 kb (ethanol - untreated) from ChEC-seq2 with Rpb1-MN, Mintbody-MNase constructs (α-Ser5p -MN, α-Ser2p-MN), Kin28-MN, and Ctk1-MN. Data were smoothed with a step size of 5 and window of 10. (**C**) Metagene plots showing the average change in CPMn (ethanol - untreated) from ChEC-seq2 of the top ethanol-induced genes (100 genes, right; *Supplementary file 1*) and the downregulated ribosomal protein genes (137 genes, left; *Supplementary file 1*). CPMn for soluble micrococcal nuclease (sMNase) is shown in black. (**D**) Gene-region enrichment of each protein relative to Rpb1-MN. For each protein and each gene, the average change in CPMn (ethanol - untreated) was calculated. Signal was binned into gene regions and normalized by region length. The length-normalized region signal relative to total signal (all gene regions) was calculated. The average for 137 RPGs and the top 100 ethanol-induced ± SEM is plotted. For all panels, the average from three biological replicates is shown.

## Developing a kinetic model for transcription based on ChEC-seq2 RNAPII occupancy

Because ChEC-seq2 provides information about important regulatory steps that have not been evident from previous global studies, we used these data (as well as ChIP-seq data) to develop models for

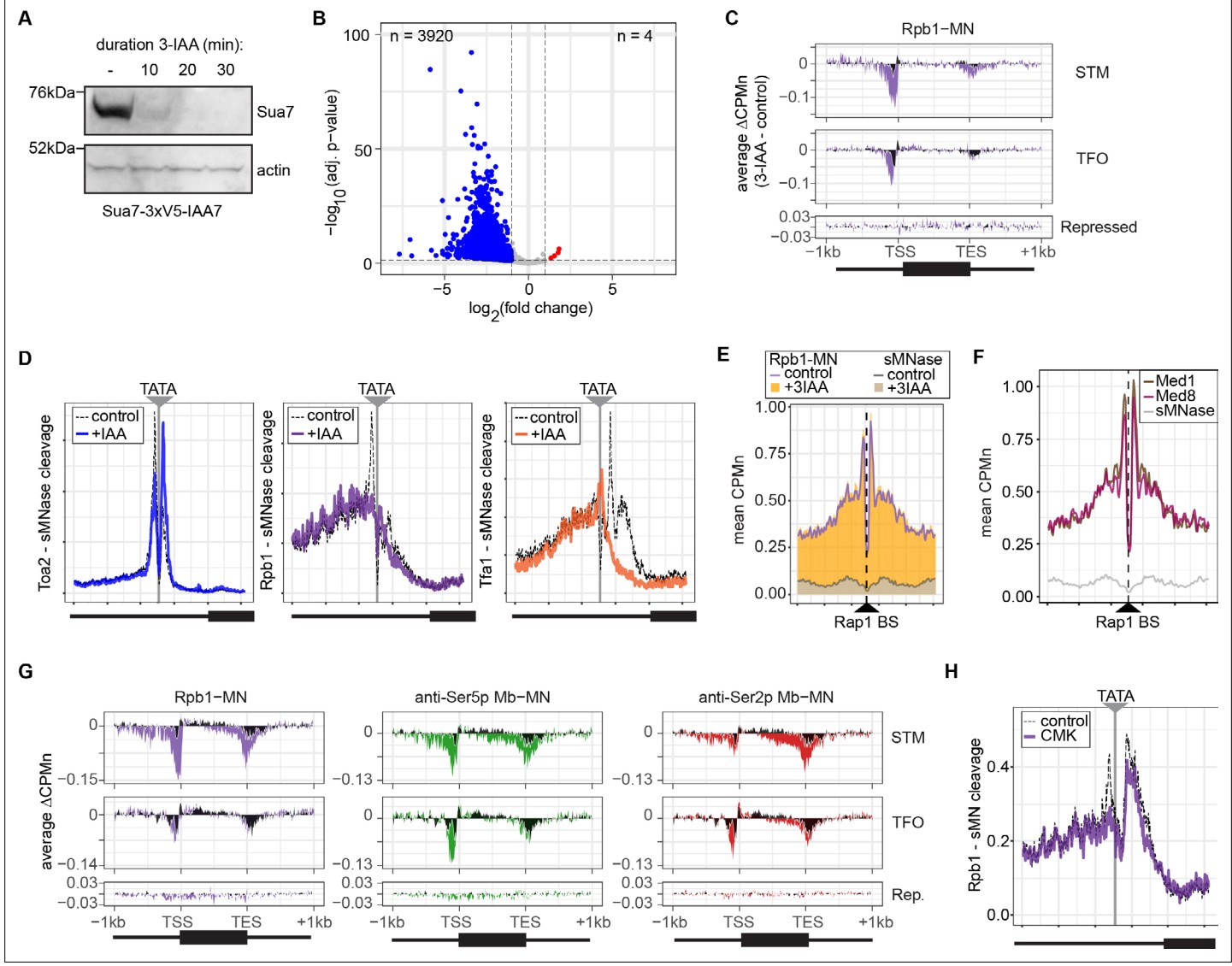

**Figure 5.** Conditional depletion of TFIIB and inhibition of TFIIH kinase cause distinct effects on promoter-associated RNA polymerase II (RNAPII). (**A**) Chemiluminescent immunoblot of Sua7-3V5-IAA7 at the indicated time points following addition of 3-IAA. Signal from actin is shown as a loading control. (**B**) Volcano plot of $\log_2$ fold-change (LFC) in nascent RNA vs. the -$\log_{10}$ of the adjusted p-value following degradation of Sua7-3V5-IAA7 via treatment with 3-IAA for 60 min. Of 5295 mRNAs, 3920 mRNAs were significantly decreased (blue; LFC ≤ –1 and adj. p<0.05) and 4 mRNAs were significantly increased (red; LFC ≥1 and adj. p<0.05). Cells were grown in synthetic complete medium. (**C**) Metagene plots showing the average change in CPMn (3-IAA - control) from ChEC-seq2 with Rpb1-MN upon degradation of Sua7-3V5-IAA7 for 20 min over 150 genes with highest expression from STM and TFO classes (**Rossi et al., 2021**) and 84 repressed genes are plotted (**Supplementary file 1**). Cells were grown in SDC. (**D**) Metasite plot over the TATA boxes ±250 bp from 597 expressed, mRNA-encoding genes (**Supplementary file 1**; **Rhee and Pugh, 2012**). In each case, soluble micrococcal nuclease (sMNase) CPMn was subtracted from the specific CPMn and the untreated control is shown in gray for comparison. The location of the TATA sequence is indicated with a gray bar. Cells were grown in YPD and Sua7-3V5-IAA7 was depleted for 60 min. (**E**) Metasite plot of average CPMn from Rpb1-MN and sMNase cleavage over 896 high-confidence Rap1 sites (**VanBelzen et al., 2024**). Purple and dark gray lines represent mean Rpb1-MN and sMNase cleavage in untreated cells; orange and gray columns represent mean Rpb1-MN and sMNase cleavage upon Sua7 depletion for 60 min. (**F**) Metasite plots of average CPMn from Med1-MN (brown), Med8-MN (magenta), sMNase (gray) over Rap1 sites as in (**E**). (**G**) Metagene plots as in (**C**) of average change in signal (CMK - control) upon inhibition of *kin28is* for Rpb1-MN (purple), Ser5p-MN (green), Ser2p-MN (red). Cells were grown in SDC and treated with 5 µM CMK for 60 min. (**H**) Metasite plot over TATA boxes as in (**D**) for the sMNase-corrected signal from Rpb1-MN before (gray) and after (purple) inhibition of *kin28-is* with 5 µM CMK for 60 min. For all panels, the average of three biological replicates is plotted.

The online version of this article includes the following source data and figure supplement(s) for figure 5:

**Source data 1.** PDF file containing labeled western blot membrane corresponding to **Figure 5A**.

**Source data 2.** Original western blot membrane corresponding to **Figure 5A**.

**Figure supplement 1.** Growth effect of CMK treatment in wild-type and *kin28is* cells.

the global kinetics of yeast RNAPII transcription. Steady-state occupancy of RNAPII should reflect the rates of several steps: RNAPII recruitment to the UAS and/or promoter, PIC assembly, initiation, elongation, and termination. We developed a stochastic computational model for these steps (*Figure 6A*) by fixing rates that have been experimentally determined ($k_1$, $k_{-1}$, $k_3$, $k_5$, $k_6$, $k_7$; *Table 1*) and optimizing the remaining rates to fit to the RNAPII occupancy observed from either ChIP-seq or ChEC-seq2. To capture the distinct mechanisms of RNAPII recruitment, we modeled the STM and TFO gene classes separately: for the STM class, we assumed that all RNAPII is recruited first to the UAS (reflecting $k_1$) before being transferred to the promoter (reflecting $k_2$); for the TFO class, RNAPII is recruited directly to the promoter (reflecting $k_3$). In genes with a UAS (i.e. STM genes), RNAPII is recruited nearly exclusively to the UAS through ssTFs and coactivators (*Baek et al., 2021*), and we therefore omitted RNAPII recruitment to the promoter ($k_3$) in the STM model. We also modeled dissociation from the UAS (reflecting $k_{-1}$, STM class) and promoter (reflecting $k_{-3}$, both classes), as well as the possibility of reversal from promoter to UAS (reflecting $k_{-2}$, STM class).

Fitting to the RNAPII occupancy from ChIP-seq or ChEC-seq2 over different regions (UAS, promoter, transcribed region, or 3'UTR), we identified the optimal range of values for the undefined rates (i.e. $k_2$, $k_{-2}$, $k_{-3}$, and $k_4$), producing an ensemble of best-fit models (*Figure 6—figure supplement 1*). Agreement between the models and the data was measured using cosine similarity (Methods). The models trained on the ChEC-seq2 occupancy for either the TFO or STM genes showed excellent agreement with the data (cosine similarity >0.995; *Figure 6B*, top and *Figure 6—figure supplement 1A and C*). Optimal agreement between the models and ChEC-seq2 data was achieved by using the lower bound for dwell time at the terminator from *Zenklusen et al., 2008*, and *Larson et al., 2011* (30 s; $k_7$=0.0325 s⁻¹; *Table 1*). Importantly, the rates that are shared between the STM and TFO models are identical (*Table 1*). Thus, modeling RNAPII occupancy data from ChEC-seq2 produced a range of plausible values for the rates of transcription that agrees well with the empirical data (*Table 1*).

Using the published rates, neither model was able to find rates for the other steps that produced occupancy that matched that observed by ChIP-seq (i.e. there were no models with cosine similarity >0.9; *Figure 6—figure supplement 1B and D*). The best ChIP-seq models predicted RNAPII occupancy over all regions that was significantly different from that observed (*Figure 6B*, bottom). By varying the published rates, the model could produce the occupancies observed by ChIP-seq (*Figure 6—figure supplement 1E and F*). However, this required eliminating dissociation from the promoter ($k_{-3}$), increasing the initiation rate ($k_5$) 2-fold with instantaneous recruitment of TFIIH ($k_4$) and increasing the termination rate ~4.3-fold above the maximum published rate ($k_7$=0.14 s⁻¹; *Figure 6—figure supplement 1E*, inset table). Thus, although it is possible to model the RNAPII occupancy observed by ChIP-seq, the predicted rates are difficult to reconcile with the literature.

We explored which rates in the model could account for the effects of TFIIB depletion (*Figure 6C*) and Kin28 inhibition (*Figure 6D*; Methods) on mean RNAPII occupancy over UASs, promoters, transcribed regions, and 3'UTRs. Consistent with a role for TFIIB in recruiting RNAPII to the promoter, reducing the rate of RNAPII recruitment ($k_3$) to the promoters of TFO genes produced RNAPII occupancy changes that matched the observed effects of TFIIB depletion (*Figure 6C*, left; *Table 1*).

For the STM genes, decreasing $k_2$ alone (i.e. the rate of RNAPII transfer from the UAS to promoter) predicted an accumulation of RNAPII at the UAS and did not agree well with the data (*Figure 6C*, right). Instead, models that decreased $k_2$ and *either* increased the rate of dissociation from the UAS ($k_{-1}$) or decreased the rate of RNAPII recruitment to the UAS ($k_1$) produced RNAPII occupancies that agreed well with the data (*Figure 6C*, right; *Table 1*). Therefore, for STM genes, the model predicts that depletion of TFIIB may both reduce RNAPII recruitment to the promoter and reduce recruitment of RNAPII to, or stimulate RNAPII dissociation from, the UAS.

Next, we asked which rates in our kinetic model could account for the effects of inhibiting Kin28. Modeling a decrease in the rate of initiation ($k_5$) predicted an accumulation at the promoter (and UASs of STM class genes), which is not observed (*Figure 6D*). Instead, the effects of inhibiting Kin28 fit best with destabilizing RNAPII bound to the UAS or promoter, either by decreasing recruitment ($k_1$ or $k_3$, respectively) or by increasing dissociation ($k_{-1}$ or $k_{-3}$, respectively; *Table 1*). Indeed, for TFO-class genes, either an increase in promoter dissociation ($k_{-3}$) or a decrease in promoter recruitment ($k_3$) with a decrease in initiation ($k_5$) produced occupancies that agreed with the data (*Figure 6D*, left; *Table 1*). Similarly, for STM class genes, incorporating an increase in promoter dissociation ($k_{-3}$), an increase in UAS dissociation ($k_{-1}$), or decrease in UAS recruitment ($k_1$) with a decrease in initiation ($k_5$) resulted in

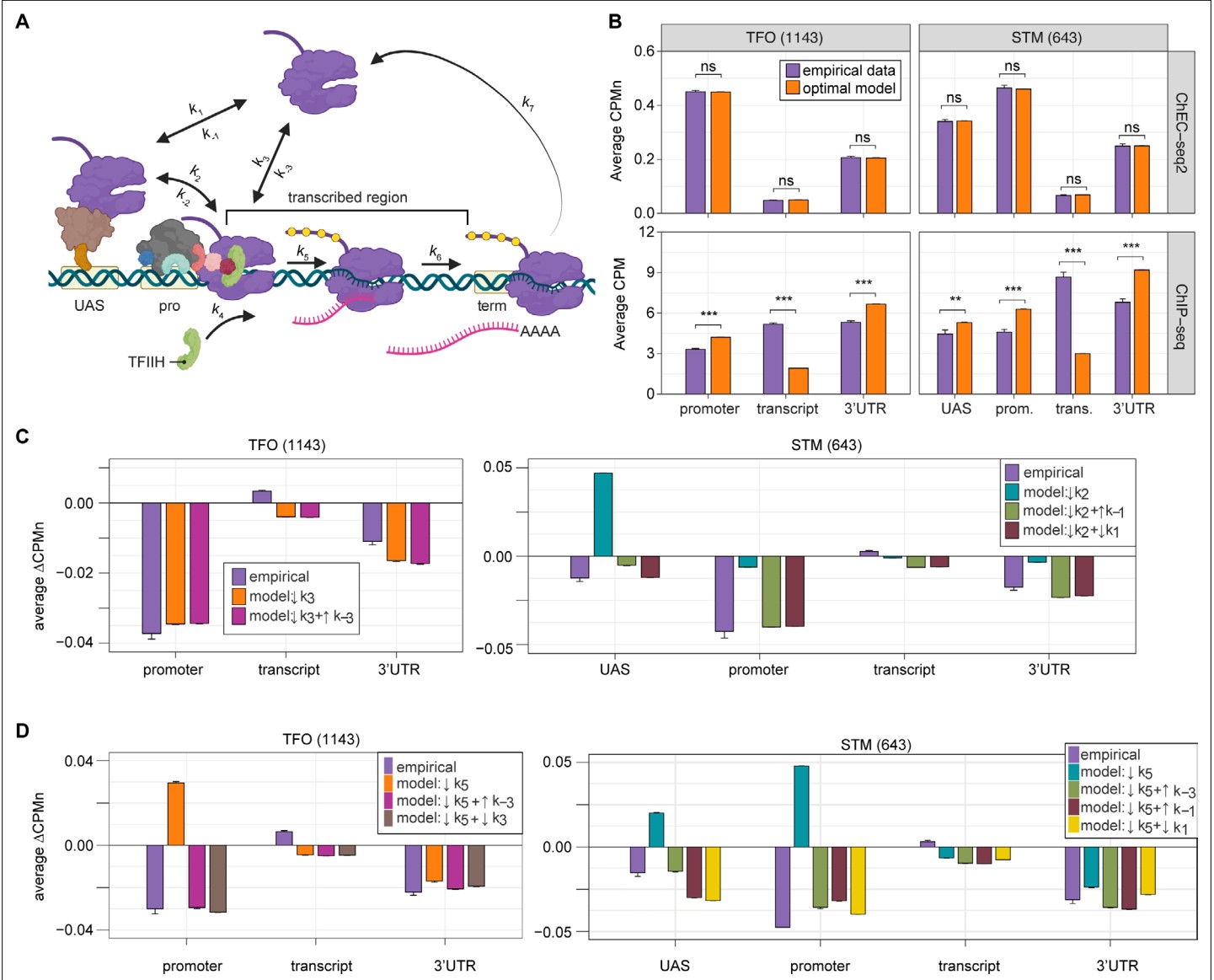

**Figure 6.** Global kinetic model for RNA polymerase II (RNAPII) transcription. (**A**) Schematic for a model of the global kinetics of transcription by RNAPII in *S. cerevisiae*. Two alternative mechanisms of RNAPII recruitment are shown: (1) direct recruitment to TFO promoters governed by rates $k_3$ and $k_{-3}$ and (2) recruitment to the STM upstream activating sequences (UASs) facilitated by a sequence-specific transcription factors (ssTFs) and coactivators such as Mediator ($k_1$ and $k_{-1}$), followed by transfer to the promoters ($k_2$ and $k_{-2}$). After RNAPII arrives at the promoter it can dissociate at rate $k_3$ until TFIIH is recruited ($k_4$), followed by initiation ($k_5$). RNAPII elongation ($k_6$) across the transcribed region produces mRNA. Pausing during termination is determined by the dissociation rate $k_7$. Transcription is modeled as a stochastic, processive process with successful recruitment of TFIIH representing the committed step. Processes involving rates $k_4$, $k_5$, $k_6$, and $k_7$ are irreversible. (**B**) The average Rpb1 signal (purple) from ChEC-seq2 (top) and ChIP-seq (bottom) over the indicated regions from 1143 TFO-class genes and 643 STM-class genes that are expressed in SDC (**Supplementary file 1**). Rates $k_2$, $k_{-2}$, $k_{-3}$, and $k_4$ were explored to fit the empirical data for each dataset. The remaining rates were drawn from published values (see **Table 1**). RNAPII occupancy was simulated across gene regions. UAS, promoter, and 3'UTR were represented by a single 120 bp bin and the transcript region was composed of 10 sequential bins to represent a 1200 bp transcript. The average predicted occupancy for RNAPII over each region from the models (i.e. sets of rates) that best matched the empirical data are shown (see Methods). For Rpb1 ChEC-seq2, 789 STM models and 371 TFO models fit the empirical data. Using the same fit thresholds for ChIP-seq data produced no models. Instead, the average predicted occupancy from an equal number of the top-performing ChIP-seq models as used in ChEC-seq2 simulations (i.e. 789 for STM and 371 for TFO) was used to generate the predictions shown (**Table 1**). (**C, D**) The average change in Rpb1-MN by ChEC-seq2 at each gene region following conditional depletion or inactivation of preinitiation complex (PIC) components (purple) for 1143 TFO-class genes and 643 STM-class genes that are expressed in SDC is shown. The rates from the ensemble of best models in (**B**) were adjusted to model the observed changes in Rpb1-MN over each gene region. (**C**) The average change in Rpb1-MN (3-IAA - control) following conditional depletion of TFIIB (purple, from **Figure 5C**). For TFO-class genes, a decrease in promoter recruitment ($\downarrow k_3$) with or without an increase in promoter dissociation ($\downarrow k_3 + \uparrow k_{-3}$) fit the empirical findings. For STM genes, a decrease in transfer from UAS ($\downarrow k_2$) combined with an increase

*Figure 6 continued on next page*

*Figure 6 continued*

in dissociation from UAS ($\downarrow k_2 + \uparrow k_{-1}$) or decrease in UAS recruitment ($\downarrow k_2 + \downarrow k_1$) fit the observed changes (*Table 2*). (**D**) The average change in Rpb1-MN (CMK - control) following inhibition of TFIIH kinase (purple, from *Figure 5G*). For TFO-class genes, a decrease in initiation ($\downarrow k_5$) combined with an increase in dissociation from promoter ($\downarrow k_5 + \uparrow k_{-3}$) or decrease in promoter recruitment ($\downarrow k_5 + \downarrow k_3$) fit the observed changes. For STM, a decrease in initiation ($\downarrow k_5$) combined with either an increase in dissociation from promoter ($\downarrow k_5 + \uparrow k_{-3}$), an increase in dissociation from UAS ($\downarrow k_5 + \uparrow k_{-1}$), or a decrease in UAS recruitment ($\downarrow k_5 + \downarrow k_1$) fit the observed changes (*Table 2*). Panel A was created with BioRender.com.

The online version of this article includes the following figure supplement(s) for figure 6:

**Figure supplement 1.** Parameter fitting of unknown transcription rates.

fits that agreed with the empirical findings (*Figure 6D*, right). Notably, for STM class genes, the combination of a decrease in initiation with an increase in promoter dissociation produced the best fit at the UAS. This suggests a feedback mechanism between PIC formation and initiation. Together, these findings indicate that the changes in RNAPII occupancy observed by ChEC-seq2 upon perturbation of PIC components can be explained by reasonable changes in transcriptional rates.

## ChEC-seq2 suggests a role for the NPC in stabilizing promoter association of RNAPII

Hundreds of active yeast genes physically associate with the NPC and this is dependent on ssTFs (*Ahmed et al., 2010*; *Brickner et al., 2019*; *Brickner et al., 2012*; *Casolari et al., 2005*; *Casolari et al., 2004*; *Light et al., 2010*; *Randise-Hinchliff et al., 2016*; *Van de Vosse et al., 2013*). Mutations that disrupt this interaction cause a quantitative decrease in transcription (*Ahmed et al., 2010*; *Brickner et al., 2016*). For example, a mutation in the Gcn4 TF that blocks interaction with the

**Table 1.** Kinetic parameters from stochastic model.

| Rate | Published Values | Standard Growth - ChEC-seq | | Standard Growth - ChIP-seq | | GCN4 | | Rate Description |
|---|---|---|---|---|---|---|---|---|
| | | Selected Rate | Functional Range | Selected Rate | Functional Range | Selected Rate | Functional Range | |
| $k_{1, UAS}$ | 0.0019–0.0027 /s (*Rosen et al., 2020*) | 0.002 /s | - | 0.002 /s | - | 0.002 /s | - | Recruitment of RNAPII to UAS |
| $k_{-1, UAS}$ | 0.001–0.005 /s (*Rosen et al., 2020*) | 0.003 /s | - | 0.003 /s | - | 0.003 /s | - | Dissociation of RNAPII from UAS |
| $k_{2, UAS}$ | None Identified | - | 0.03–0.1 /s | - | 0.03–0.1 /s | - | 0.03–0.1 /s | Transfer of RNAPII from UAS to Promoter. Requirement for assembly early PIC components at Promoter is incorporated |
| $k_{-2, UAS}$ | None Identified | - | 0.0–0.07 /s | - | 0.0–0.1 /s | - | 0.0–0.04 /s | Transfer of RNAPII from Promoter to UAS |
| $k_{3, Promoter}$ | None identified. Used $k_1$ | 0.002 /s | - | 0.002 /s | - | NA | - | Recruitment of RNAPII to Promoter |
| $k_{-3}$ | None Identified | - | 0.0–0.03 /s | - | 0.0–0.03 /s | - | 0.0–0.03 /s | Dissociation of RNAPII from Promoter |
| $k_4$ | None Identified | - | 0.0075–0.0275 /s | - | 0.025–0.1 /s | - | 0.01–0.03/s | Recruitment of TFIIH |
| $k_5$ | 0.1 /s (TFIIH residency, *Nguyen et al., 2021*) | 0.1 /s | - | 0.1 /s | - | 0.1 /s | - | Initiation and phosphorylation of RNAPII CTD by TFIIH Kinase (Kin28) |
| $k_6$ | 1–3 kb/min | 1 kb/min | - | 1 kb/min | - | 1 kb/min | - | Transcription elongation |
| $k_7$ | 0.009–0.034/s (*Larson et al., 2011* on MDN1) | 0.0325 /s | - | 0.0325 /s | - | 0.0325 /s | - | Terminantion pause |

NPC results in a quantitative decrease in transcription of Gcn4 targets (genes involved in amino acid biosynthesis; *Brickner et al., 2019*; *Hinnebusch and Fink, 1983*). This mutation replaces three amino acids within a 27 amino acid positioning domain (PD$_{GCN4}$) that does not overlap the activation or DNA binding domains (*Brickner et al., 2019*). We confirmed this effect by measuring nascent transcription upon amino acid starvation in *gcn4-pd* strains or a wild-type control (Methods). Although both *GCN4* and *gcn4-pd* mutant strains showed widespread transcriptional changes upon amino acid starvation (*Figure 7A*), the upregulation (and downregulation) of transcription was quantitatively stronger for the *GCN4* strain (*Figure 7A*, right panel). We tested if this transcriptional defect is associated with a competitive fitness defect by competing *GCN4* and *gcn4-pd* strains in the absence of histidine±3-amino triazole (3-AT, an inhibitor of the His3 enzyme, which selects for maximal expression of *HIS3*). The relative abundance of *GCN4* and *gcn4-pd* strains was quantified using Sanger sequencing (*Sump et al., 2022*). The *GCN4* strain showed greater fitness under both conditions, but this was particularly evident in the presence of 3-AT (*Figure 7B*).

ChEC-seq2 against Rpb1-MN was performed in *GCN4*, *gcn4Δ*, and *gcn4-pd* mutant strains grown in the presence or absence of amino acids. This experiment identified 287 genes that showed a log$_2$ fold-change of 1 or greater (adj. p<0.05) in the *GCN4* strain upon amino acid starvation, but not in the *gcn4Δ* strain (*Supplementary file 1*). These genes were strongly enriched for genes involved in amino acid metabolism (p=3e-46; GO term 0006520) and strongly overlapped with Gcn4 targets (Bonferroni-adjusted p=1e-10 from Fisher's exact test comparing overlap with targets defined near high-confidence Gcn4 ChEC-seq2 sites; *VanBelzen et al., 2024*). In the presence of amino acids, neither the *gcn4Δ* nor *gcn4-pd* mutations affected Rpb1 occupancy at the 287 Gcn4-dependent genes (*Figure 7C*, left column). However, upon amino acid starvation, strains lacking Gcn4 showed a stark decrease in Rpb1 recruitment upstream of the TSS that spanned both the UAS and promoter region (*Figure 7C*, top panel). The *gcn4-pd* mutation resulted in a more modest decrease in Rpb1 specifically over the promoter (*Figure 7C*, bottom). This suggested that the recruitment of RNAPII to the UAS region is dependent on Gcn4, but not on the PD$_{GCN4}$.

Consistent with this possibility, Rpb1 cleavage adjacent to the TATA boxes near the Gcn4 target genes and over Gcn4 binding sites was strongly decreased by loss of Gcn4 (p<2e-16; Kolmogorov-Smirnov test comparing the mean cleavage pattern over 173 TATAs; *Figure 7D*, right panels). In the *gcn4-pd* strain, Rpb1 cleavage was decreased over the TATA box (p=4e-5; *Figure 7D*, top right), but not over the Gcn4 binding site (*Figure 7D*, bottom right). Thus, the PD$_{GCN4}$ promotes the association of RNAPII with promoters, but does not affect RNAPII binding to the UAS.

Finally, we compared the effects of adjusting the rates of each step in our kinetic model to the effects of the Gcn4 mutations on Rpb1 occupancy (*Figure 7—figure supplement 1*). The effects of loss of Gcn4 agreed well with simply decreasing RNAPII recruitment to the UAS alone ($k_1$, resulting in less RNAPII to move from the UAS to the promoter; *Figure 7E*). For the *gcn4-pd* mutant, increasing the dissociation of RNAPII from the promoter ($k_{-3}$) either alone or in combination with decreasing the rate of transfer of RNAPII from the UAS to the promoter ($k_2$) agreed well with these data (*Figure 7E*). This suggests that Gcn4 recruits RNAPII to the UAS through its activation domains and that its interaction with the NPC stabilizes promoter-bound RNAPII.

## Discussion

Understanding complex biological mechanisms requires multipronged, multidisciplinary approaches. Each approach has strengths and weaknesses but together, they provide a more complete picture. Our current understanding of RNAPII transcription, involving the dynamic collaboration of dozens of proteins, is the product of biochemical, structural, genetic, cell biological, and genomic approaches. From decades of such work, we have an excellent working model for this critical biological process. Biochemical, structural, and cell biological approaches (and, in some cases, genetic approaches) can be biased by the particularities of the model system(s). For this reason, global approaches provide an essential perspective to assess the generality of the conclusions from more focused studies. Our current global perspective of molecular biology is dominated by a single technique: ChIP-seq and its derivatives. Indeed, ChIP-seq is the sole method used to define DNA binding and chromatin state by the ENCODE and modENCODE Consortium (*Landt et al., 2012*). Such a methodological monoculture is problematic if there are ways in which ChIP falters in detecting important interactions (*Park et al., 2013*; *Teytelman et al., 2013*).

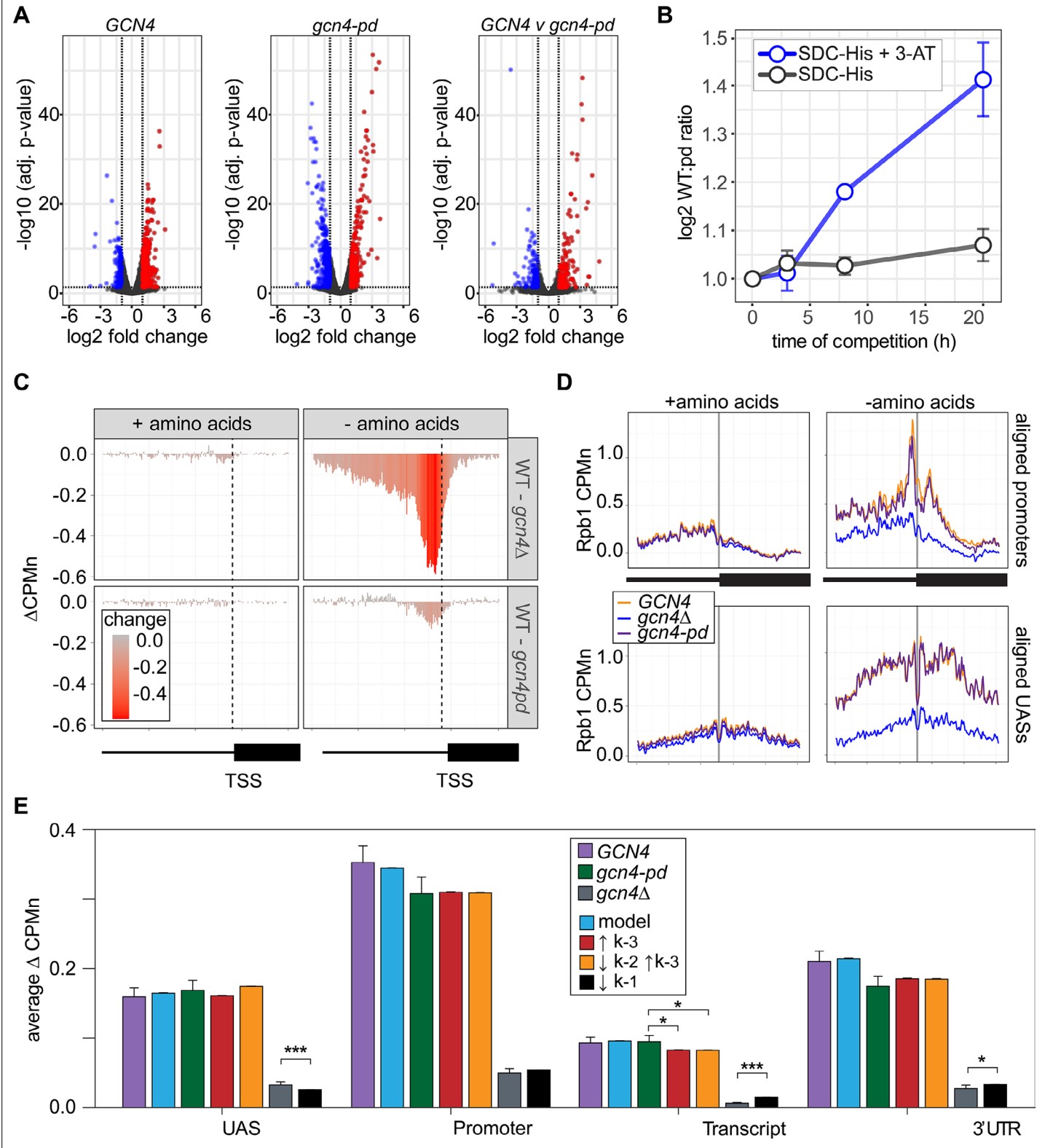

**Figure 7.** The Gcn4 positioning domain stabilizes RNAPII association with the promoter without affecting recruitment to the upstream activating sequence (UAS). (**A**) Volcano plot of $\log_2$ fold-change in nascent RNA vs. $-\log_{10}$ adjusted p-values from cells starved for histidine for 1 hr vs. cells in complete medium. Nascent RNA counts for a total of 5295 mRNAs were measured in *GCN4* (left) and *gcn4-pd* (middle). The $\log_2$ fold-change in nascent RNA vs. $-\log_{10}$ adjusted p-values in *GCN4* vs. *gcn4-pd* from cells grown in the absence of histidine (right). Significantly downregulated (blue; LFC ≤ −1 and adj. p<0.05), upregulated (red; LFC ≥1 and adj. p<0.05) genes are highlighted. (**B**) Relative abundance of *GCN4* and *gcn4-pd* strains in a

*Figure 7 continued on next page*

*Figure 7 continued*

mixed culture, determined by quantifying the relative abundance of the two alleles in the population (*Sump et al., 2022*) in either SDC-His or SDC-His+10 mM 3-amino triazole (3-AT). (**C**) Average difference in Rpb1-MN between *gcn4Δ* (top) and *gcn4-pd* (bottom; mutant - *GCN4*, ΔCPMn) upstream of 246 Gcn4-target genes for cells grown + amino acids (left) or - amino acids for 1 hr (*Supplementary file 1*). A region spanning 700 bp upstream and 400 bp downstream of the transcription start site (TSS) (hashed vertical line) is displayed and the color scale reflects the difference between wild-type and each mutant. (**D**) Metasite plot showing average Rpb1-MN CPMn from cells grown + amino acids (left) or - amino acids (right) for *gcn4Δ* (blue), *gcn4-pd* (purple), and *GCN4* (yellow) strains. Top: Rpb1-MN-sMNase CPMn over TATA boxes±250 bp upstream of 173 Gcn4-dependent genes (*Supplementary file 1*; *Rhee and Pugh, 2012*). Bottom: Rpb1-MN CPMn over 284 Gcn4 binding sites (Gcn4BS)±250 bp upstream of 130 Gcn4 target genes (*Supplementary file 1*). (**E**) The average change in Rpb1-MN by ChEC-seq2 at each gene region in cells shifted into media lacking amino acids (SD+uracil) vs. cells shifted into media with amino acids (SDC) is plotted (SD+uracil - SDC, ΔCPMn) for each strain (*GCN4*, purple; *gcn4-pd*, green; *gcn4Δ*, gray). Rates $k_2$, $k_{-2}$, $k_{-3}$, and $k_4$ were re-fit to the observed Rpb1-MN occupancy at 287 Gcn4-dependent genes in wild-type cells grown in the absence of amino acids and yielded 1057 STM models (blue). The rates from these best-fit models were adjusted to fit the observed changes in Rpb1-MN over each gene region in *gcn4-pd* and *gcn4Δ* strains. For *gcn4-pd*, an increase in dissociation from the promoter fit the empirical findings (↑$k_{-3}$, red; *Table 2*) or a combined increase in promoter dissociation and decrease in transfer from UAS to promoter (↓$k_{-2}$↑$k_{-3}$, orange; *Table 2*). For *gcn4Δ*, a decrease in recruitment to UAS fit in the model and the empirical findings (↓$k_1$, black; *Table 2*).

The online version of this article includes the following figure supplement(s) for figure 7:

**Figure supplement 1.** Parameter fitting of unknown transcription rates in upstream activating sequence (UAS)-recruitment model for Gcn4 target genes.

## ChIP vs. ChEC

For proteins that bind directly to DNA at specific sites, ChIP-seq and ChEC-seq2 generally agree. For example, high-confidence binding sites for ssTFs show excellent agreement between either ChIP-seq or ChIP-exo and ChEC-seq (*Donczew et al., 2021*; *VanBelzen et al., 2024*; *Zentner et al., 2021*; *Zentner et al., 2015*). Likewise, mapping the associations of PIC components, Mediator or the kinases associated with transcription by ChEC-seq2 was very similar to such maps produced by ChIP-seq (*Saleh et al., 2021*; *Wong et al., 2014*). However, some exceptions have been noted, as well. ChEC-seq with both Rif1 and Sfp1 reveals biologically sensible binding sites that were not evident from ChIP-seq (*Bruzzone et al., 2021*).

While both ChIP-seq and ChEC-seq2 with RNAPII gives enrichment over genes that correlates with transcription, the patterns are complementary; ChIP highlights interactions with the transcribed region (reflecting paused or elongating RNAPII) and ChEC highlights interactions with the enhancer, promoter, and terminator (reflecting preinitiation or terminating RNAPII). We have validated the RNAPII enrichment reported by ChEC-seq2 in five ways. First, the maps produced by two different subunits of RNAPII are highly similar (*Figure 2*). Second, the RNAPII ChEC-seq2 signal over promoters and UAS regions correlates well with the RNAPII ChIP signal over coding sequences and with nascent transcription rates (measured by SLAM-seq; *Figure 2*). Third, the cleavage by either Rpb1 or Rpb3 (as well as TFIIA and TFIIE) peaks on either side of TATA boxes, which agrees well with biochemical and structural analysis of the PIC (*Figure 2*, *Video 1*). Fourth, widespread changes in transcription are captured by changes in the Rpb1 enrichment by ChEC-seq2 over all gene regions (*Figure 4*). Fifth, depletion of TFIIB leads to loss of Rpb1 and TFIIE, as well as an increase of TFIIA, over TSSs (*Figure 5*). These data, strengthened by the correlations with ChEC-seq2 using factors involved in initiation and elongation, argue that the patterns of RNAPII enrichment revealed by ChEC-seq2 are biologically meaningful and fit well with the literature.

Why is there a difference between RNAPII ChIP-seq and ChEC-seq2? While ChIP captures direct protein-DNA interactions well, it is much less able to capture indirect interactions. Additional factors that may influence ChIP enrichment include the local nucleosome occupancy, the accessibility of the epitope, and the relative sensitivity of different regions to shearing by sonication (*Giresi et al., 2007*). Unlike ssTFs or even PIC components that bind to directly to precise genomic sites, RNAPII interacts both indirectly (through ssTFs) and directly (in the PIC and transcribing RNAPII) with different regions, each of which is associated with distinct sets of cofactors. These differences likely impact the two methods; ChEC should detect both direct and indirect interactions with DNA, whereas ChIP should strongly favor direct interactions. Likewise, ChEC will perform better in nucleosome-depleted regions while ChIP crosslinking may be enhanced by lysine-rich nucleosomes.

ChEC-seq2 detects UAS-associated RNAPII observed in single-molecule biochemical experiments (*Baek et al., 2021*; *Rosen et al., 2020*) that have not been observed by ChIP-seq. This is consistent

with recruitment of RNAPII to ssTFs/Mediator bound to UASs. While the enhanced RNAPII ChEC-seq2 signal in intergenic regions may also reflect lower nucleosome occupancy, sMNase cleavage was not enriched over UASs like Rap1 binding sites (*Figure 5E*). Because such sites are also occupied by Mediator (*Figure 5F*), this supports a Mediator-dependent mechanism of RNAPII recruitment to the UAS (*Baek et al., 2021*; *Rosen et al., 2020*). Furthermore, it is important to highlight that the RNAPII ChEC-seq2 enrichment observed over promoters and UASs is consistent with that expected from known dwell times and the rate of elongation (*Figure 2*). The low RNAPII ChIP-seq signal at UASs and the high signal over coding sequences could reflect both its more direct interaction with DNA and its intimate association highly crosslinkable nucleosomes during transcription (*Bintu et al., 2012*; *Ehara et al., 2022*; *Ehara et al., 2019*; *Kujirai et al., 2018*). However, it is less clear why the RNAPII ChIP-seq signal over the promoter is so low. ChIP-seq successfully captures enrichment of PIC components at promoters, indicating that promoter regions can be successfully enriched by ChIP. Future studies will resolve these differences.

## ChEC-seq2 with elongation factors

We also present a novel method for observing the genome-wide location of the phosphorylated forms of RNAPII (Ser2p and Ser5p) using single chain antibodies (Mintbodies) tagged with MNase. ChEC-seq2 with these Mintbodies produces patterns that agree well with total RNAPII and with the kinases responsible for these modifications. Consistent with ChIP, Ser5p RNAPII is enriched in promoters and the 5′ end of active genes, while Ser2p is enriched over the body and 3′ end. Inactivation of the Kin28 Ser5p kinase results in dramatic loss of RNAPII, Ser5p RNAPII, and Ser2p RNAPII from active genes (*Figure 5*). This is consistent with an important role for Ser5p in initiation and with the observation that Ser2 phosphorylation is functionally downstream of Ser5p.

ChEC-seq2 with factors involved in elongation (Ctk1, Spt5, Ser2p-RNAPII), when normalized to total RNAPII, showed greater enrichment over the CDS (*Figure 3G*), as expected. However, it is surprising that we also observed clear enrichment of these factors at promoters (e.g. *Figure 3A, E, and F*). The association of elongation factors with the promoter seems to be biologically relevant. Changes in transcription correlate with changes in ChEC-seq2 enrichment for these factors and modifications (*Figure 4C*). Blocking initiation by inhibiting TFIIH kinase led to a reduction of Ser5p RNAPII and Ser2p RNAPII over both the promoter and the transcribed region (*Figure 5G*). This suggests either that the true signal of these factors over transcribed regions is less evident by ChEC-seq2 than by ChIP-seq or that ChEC-seq2 can reveal interactions of elongation factors at early stages of transcription that are missed by ChIP-seq. The expectations for enrichment of elongation factors and phosphorylated CTD are largely based on ChIP data. Because ChIP-seq fails to capture RNAPII enrichment at UASs and promoters, it is possible that ChIP also fails to capture promoter interaction of factors involved in elongation as well.

Factors important for elongation can also function at the promoter. For example, Ctk1 is required for the dissociation of basal TFs from RNAPII at the promoter (*Ahn et al., 2009*). Transcriptional induction leads to increases in Ctk1 ChEC-seq2 enrichment both over the promoter and over the 3′ end of the transcribed region (*Figure 4C*). Dynamics of Spt4/5 association with RNAPII from in vitro imaging (*Rosen et al., 2020*) indicate that the majority of Spt4/5 binding to RNAPII does not lead to elongation; Spt4/5 frequently dissociates from DNA-bound RNAPII. Association of Spt4/5 with RNAPII may represent a slow, inefficient step in the transition to productive elongation. If so, then ChEC-seq2 may capture transient Spt4/5 interactions that occur prior to productive elongation, producing enrichment of Spt5 at the promoter.

## A role for interaction with the NPC in stabilizing the PIC

The NPC has been implicated in transcription in yeast and other organisms. In yeast, inactivation of DNA elements or TFs that promote interaction with the NPC leads to a quantitative defect in transcription (*Ahmed et al., 2010*; *Brickner et al., 2012*). Single-molecule RNA FISH (smRNA FISH) in strains bearing mutations that blocked the interaction of the *GAL1-10* promoter with the NPC showed a decrease in the fraction of cells that exhibit transcription (*Brickner et al., 2016*). A mutation in the Gcn4 ssTF that blocks its ability to mediate peripheral localization and interaction with the NPC leads to a defect in expression of Gcn4 target genes (*gcn4-pd*; *Figure 7*; *Brickner et al., 2019*) and inactivation of nuclear pore proteins essential for chromatin interaction leads to a global

transcriptional defect (*Ge et al., 2024*). Applying RNAPII ChEC-seq2, we have explored the phenotype of the *gcn4-pd* mutant. Whereas loss of Gcn4 leads to loss of RNAPII from UASs and promoters, inactivation of the PD$_{GCN4}$ reduces the association of RNAPII with the promoter without affecting its recruitment to the UAS (*Figure 7*). This suggests that the PD$_{GCN4}$ either enhances the transfer of RNAPII from the UAS to the promoter or stabilizes the association of RNAPII with the promoter. Genetic interactions between nuclear pore proteins and Mediator suggest that these two components function at the same step in transcription (*Ge et al., 2024*). Together with the smRNA FISH result, this suggests that nuclear pore proteins stimulate enhancer function by stabilizing RNAPII association with the PIC.

## A global model for yeast RNAPII kinetics

Because ChEC-seq2 measures global occupancy of RNAPII that includes important states that are missed by ChIP-seq, it allowed us to develop a global model for the kinetics of RNAPII transcription. Building on previous work (*Rossi et al., 2021*), we have modeled two classes of genes: those that show RNAPII association only with promoters (TFO) and those that show association with UASs as well (STM). For the TFO model, RNAPII is recruited directly to the promoter. For STM genes, RNAPII is recruited to the UAS and then transferred to the promoter. Subsequent steps (initiation, elongation, and termination) are assumed to be the same between these two classes. Several of the rates are from the literature, while the others were fit to the experimental RNAPII enrichments over UASs, promoters, transcribed regions, and 3'UTRs. While we were unable to find rates within a reasonable range of parameters that produced RNAPII occupancies matching ChIP-seq, the model identified a large ensemble of rates that produced RNAPII occupancies matching ChEC-seq2 (*Figure 6B*). The RNAPII occupancy from ChEC-seq2 data over highly active genes matched models that included a short dwell time over the terminator (~30 s), at the lower bound of what was reported in *Zenklusen et al., 2008* (mean = 56 ± 20 s) and *Larson et al., 2011* (mean = 70 ± 41 s).

The kinetic model suggests that perturbations often have more than one effect, as expected for a dynamic, multi-step process like transcription. For example, the effects of depletion of TFIIB on RNAPII ChEC-seq2 are best modeled by both a decrease in RNAPII recruitment and an increase in non-productive dissociation of RNAPII, either from the promoter or the UAS (*Figure 6C*). Likewise, the effects of inhibition of Kin28 were most consistent with both a decrease in initiation and an increase in dissociation from the promoter/UAS (*Figure 6D*). These results suggest that the PIC is unstable and that such perturbations cause RNAPII to dissociate. This conclusion agrees with the observation that a small fraction of the polymerases that assemble at the promoter initiate transcription (*Darzacq et al., 2007*) and with the observation that conditional inactivation of PIC components does not preserve stable intermediates (*Petrenko et al., 2019*). Moreover, these results were consistent across the entire ensemble of models, showing that this is a robust effect. These models should serve as a helpful framework for future global studies of transcription.

## Methods

### Yeast strains

Yeast strains and tagging vectors used in this study are provided in *Supplementary files 2 and 3*. C-terminal MNase fusions were introduced by recombination as previously described (*VanBelzen et al., 2024*). Sua7 was tagged with 3xV5-IAA7 using pV5-IAA7-His3MX6, which was generated by swapping the His3MX6 marker in place of the HIS3 marker in pGZ363 (*Tourigny et al., 2021*). OsTir1-LEU2 was PCR amplified from pSB2271 (*Miller et al., 2016*) with primers that facilitated recombination at *leu2Δ0* and simultaneously restored the locus to *LEU2*. The *kin28is* mutations V21C and L83G (*Rodríguez-Molina et al., 2016*) were introduced by two subsequent rounds of CRISPR-Cas9-mediated mutagenesis as described (*Anand et al., 2017*). The *GCN4-sm* and *gcn4-pd* mutations were introduced by CRISPR-Cas9-mediated mutagenesis and are described (*Ge et al., 2024*).

Mb-MN constructs were synthesized by Integrated DNA Technologies as gBlocks. The gBlocks were flanked by a *Hin*dIII and *Bam*HI site, which were used to clone the gBlocks into the pFA6a-NatMX6 vector (*Hentges et al., 2005*). The constructs were amplified from plasmid by PCR to yield amplicons flanked with homology to the *his3Δ1* locus, which were then transformed into yeast.

Strains were confirmed to have the desired sequence by amplifying the modified locus from genomic DNA and sequencing. Platinum SuperFi (Thermo Fisher Scientific) was used to amplify long targets by PCR.

## Media and growth conditions

Media were prepared as described (*Burke et al., 2000*). Cells were grown at 30°C with shaking at 200 rpm in SDC media. YPD media was used in growth assays and in *Figure 2A-C*, *Figure 2—figure supplement 2*, where cells were grown in YPD to match conditions of ChIP-seq samples. Ethanol stress was induced by growing cells in media spiked with 10% ethanol for 1 hr. Sua7-IAA7 was degraded for by treating cells with 0.5 mM indole-3-acetic acid for 60 min in SLAM-seq experiments or 20 min in ChEC-seq2 experiments. For Kin28 inhibition experiments, cells harboring the *kin28is* mutation were treated with 5 μM CMK for 60 min.

For SLAM-seq and growth competition experiments with *GCN4-sm* and *gcn4-pd*, cells were grown in SDC and then shifted into SDC or SDC-His for 1 hr. Growth competition assays were performed as described (*Sump et al., 2022*) and the histidine synthesis pathway was blocked through the addition of 3-AT to the media. For ChEC-seq2 experiments with *GCN4-sm*, *gcn4-pd*, and *gcn4Δ*, cells were grown in YPD before shifting into either SDC or SD+uracil for 1 hr.

## ChEC-seq2

The ChEC-seq2 method was performed as described (*VanBelzen et al., 2024*). Cells were permeabilized and 2 mM calcium was added to activate MNase activity. Reactions were stopped after genomic DNA was partially digested (e.g. *Figure 3—figure supplement 1A*), DNA was purified, DNA ends were repaired and ligated to an Illumina-compatible adapter (*VanBelzen et al., 2024*). A second adapter was incorporated through Tn5-based Tagmentation. Complete adapters and library indexes were incorporated through library amplification with Nextera XT Index Primers.

## ChIP-seq

Cell fixation and chromatin isolation was performed as previously described (*Kuo and Allis, 1999*) but is briefly described here for clarity. Independent cultures of BY4741 (Rpb1) and JVY022 (Rpb1-MN) were grown in YPD at 30°C, 200 rpm until cultures reached a density between 0.6 and 0.9 (OD$_{600}$). A culture volume of 100 ml was crosslinked with 1% formaldehyde for 10 min at 30°C with gentle mixing. The crosslinking reaction was quenched with 0.3 M glycine for 5 min at 30°C with gentle mixing. A volume of 50 ml was collected by centrifugation and the pellet was washed twice in ice-cold Tris-buffered saline (20 mM Tris-HCl, pH 7.5; 150 mM NaCl), snap-frozen in liquid nitrogen, and stored at –80°C for up to 2 weeks. Pellets were briefly thawed on ice and resuspended in 600 μl of ice-cold FA lysis buffer (50 mM HEPES-KOH, pH 7.5; 140 mM NaCl; 1 mM EDTA; 1% Triton X-100; 0.1% sodium deoxycholate) supplemented with protease inhibitors (1 mM PMSF; 1 μg/ml Leupeptin; 1 μg/ml Pepstatin A; 10 μg/ml Aprotinin). A volume of 600 μl zirconia beads (0.5 mm diameter) was added, and cells were lysed by bead-beating at 4°C in a Vortex Genie for 7 cycles of: 3 min on (highest setting); 1 min on ice. The lysate was separated from the beads and brought to a final volume of 600 μl with ice-cold FA lysis buffer, which was split into two 300 μl fractions for sonication. Sonication was performed on a BioRuptor Pico (Diagenode) at 4°C for 6 cycles of: 30 s on (high setting); 30 s off. Debris was pelleted by centrifugation at 17,000 × *g* for 15 min at 4°C, and the chromatin-containing supernatant was collected.

Immunoprecipitation was adapted from *Sump et al., 2022*. Dynabeads Protein G (Thermo Fisher Scientific # 10003D) were equilibrated in chilled FA lysis buffer for 2 hr at 4°C on a rotating stand. Simultaneously, 2 mg of chromatin in a 1 ml volume was incubated with 2 μl of Anti-Rpb1 (Clone 8WG16, BioLegend) at 4°C on a rotating stand. 20 μl of equilibrated Dynabeads was added to each chromatin sample and incubated overnight at 4°C on an inverting rotator. Beads were immobilized with a magnetic stand and washed four times in 1 ml of chilled Wash Buffer (50 mM HEPES-KOH, pH 7.5; 500 mM NaCl; 1 mM EDTA; 1% Triton X-100; 0.1% sodium deoxycholate) supplemented with protease inhibitors (see above). Protein of interest was eluted in 100 μl of Elution Buffer (50 mM Tris-HCl, pH 8.0; 10 mM EDTA; 1% SDS) and crosslinks were reversed by heating overnight at 65°C. Added 5 μl RNAse A (10 μg/μl) and heated at 37°C for 30 min to degrade RNA. Added 10 μl of Proteinase K (20 μg/μl) and incubated at 50°C for 1 hr. Purified DNA with QIAquick spin columns

according to the manufacturer's instructions (QIAGEN # 28104). DNA fragment size was measured on a TapeStation 4150 and confirmed to be approximately 400 bp. Sequencing libraries were prepared from 0.5 ng of DNA with the MicroPlex Library Preparation Kit v3 with dual indexes (Diagenode # C05010001 and C05010004). Libraries were sequenced at NUseq on the NovaSeq X Plus (Illumina) in the paired-end, 50 bp format. Bioinformatic analysis was performed as in ChEC-seq2 (*VanBelzen et al., 2024*), except reads were mapped with paired-end mode of Bowtie 2 (*Langmead and Salzberg, 2012*) and the ChEC-specific trimming step was omitted.

## SLAM-seq

SLAM-seq was performed as previously described (*Herzog et al., 2017*) with the following modifications. Approximately $10^8$ cells were collected, resuspended in SDC-uracil+200 µM 4-thiouracil, and incubated for 6 min at 30°C. Cells were collected by centrifugation and frozen in liquid nitrogen. RNA was extracted from cell pellets as described (*Schmitt et al., 1990*), and purified with the Monarch Total RNA Miniprep Kit (New England Biolabs). Alkylated RNA was purified with the Monarch RNA Cleanup Kit (New England Biolabs). RNA quality was confirmed after each purification with a TapeStation 4150 (Agilent). Sequencing libraries were prepared from 150 ng RNA using the QuantSeq 3' mRNA-Seq Library Prep Kit (FWD) kit (Lexogen). Sequencing was performed on a HiSeq 4000 (Illumina) in the single-end, 50 bp format at the Northwestern University NUseq core facility. In the case of SLAM-seq performed with JBY555 (*gcn4-pd-GFP*) and JBY556 (*GCN4sm-GFP*) (*Ge et al., 2024*), cells were shifted into SDC-uracil with 2 mM 4-thiouracil for 6 min.

Reads were mapped with SlamDunk (*Herzog et al., 2017*) to the S288C genome (build R64-3-1) and binned into genes classified as Verified or Uncharacterized by the Saccharomyces Genome Database. This yielded counts values for 5925 genes. Counts files were analyzed in R with DESeq2 (*Love et al., 2014*) to identify differentially expressed genes between conditions.

## Immunoblotting

Protein was isolated from cells as described (*Rüegsegger et al., 2001*) and quantified by BCA protein assay (#23225, Thermo Fisher Scientific). 40 µg of protein was separated on 10% surePAGE Bis-Tris gels in MOPS running buffer (#M00665, GenScript) and transferred to a nitrocellulose membrane. The membrane was blocked with 5% nonfat dry milk in TBST with 0.05% Tween 20 for 1 hr at room temperature and then probed with anti-V5 (#R960-25, Thermo Fisher Scientific) and anti-ß-Actin (#GTX629630, GeneTex) primary antibodies overnight at 4°C. Membranes were washed for 5 min with TBST for a total of five washes, and then incubated with goat anti-mouse conjugated with HRP (#AP127P, MilliporeSigma) in 5% milk TBST for 1 hr at room temperature. Washes were repeated and then HRP was activated with chemiluminescence reagents (#34075, Thermo Fisher Scientific) for 5 min. Blots were imaged on an c600 imaging system (Azure Biosystems).

## Computational model

We used a stochastic model to simulate the average occupancy of RNAPII along a discretized model gene (*Figure 6A*), assuming each step in the transcription cycle is a Poisson process. We separately modeled two classes of genes: STM genes and TFO genes. For STM genes, we assume that the association of RNAPII with the gene occurs at the UAS and is reversible, with association rate $k_1$ and dissociation rate $k_{-1}$. Next, the RNAPII transitions from the UAS to the promoter with rate $k_2$. This rate represents an aggregate step that requires the recruitment of early GTFs such as TFIIA and TFIIB. Because these interactions are reversible, we assume RNAPII can transition back to the UAS from the promoter with rate $k_{-2}$. When at the promoter, the RNAPII awaits the arrival of late GTFs such as TFIIH to form the complete PIC. This process occurs at the aggregate rate $k_4$. While awaiting arrival of late GTFs, the RNAPII can also dissociate from the promoter with rate $k_{-3}$. Once the PIC has assembled, TFIIH kinase phosphorylates the C-terminal domain of RNAPII to initiate transcription and promoter escape. This occurs with rate $k_5$. The transcribed region is modeled as 10 identical 120 bp compartments, and the RNAPII moves to each succeeding compartment with rate $k_6$. Finally, once the RNAPII reaches the terminator, it dissociates with rate $k_7$. TFO genes are modeled similarly, with the omission of $k_1, k_{-1}$, $k_2$, and $k_{-2}$, and instead introducing $k_3$, the rate of recruitment directly to the promoter. The UAS, promoter, and terminator regions are modeled as independent 120 bp compartments. No compartment could be occupied by more than one RNAPII.

We simulated 1000 seconds of the transcription cycle to allow the system to reach steady state. We report the RNAPII occupancy of each segment of the gene over the final 60 s to align with the experimental procedure. The simulated data was then normalized using the $L^2$ norm and scaled to have the same magnitude as the empirical data to approximate the unit conversion to CPM or CPMn. This process was repeated across 100,000 genes and the average occupancy in each region of the gene was recorded. Simulations were performed using the Gillespie algorithm (*Gillespie, 1977*), a stochastic simulation method that generates statistically correct trajectories of a given system. The algorithm uses random sampling to determine the timing and sequence of state transitions that correspond

**Table 2.** Kinetic parameters from modeling genetic perturbations of transcription.

| Experiment | Gene Class | Model # | $k_{1,UAS}$ 0.002 /s | $k_{-1,UAS}$ 0.003 /s | $k_{2,UAS}$ 0.03–0.1 /s | $k_{-2,UAS}$ 0.0–0.07 /s | $k_{3,Promoter}$ 0.002 /s | $k_{-3}$ 0.0–0.03 /s | $k_4$ 0.0075–0.0275 /s | $k_5$ 0.1 /s | $k_6$ 1 kb/min | $k_7$ 0.0325 /s |
|---|---|---|---|---|---|---|---|---|---|---|---|---|
| TFIIB Degradation, 20 min | STM | 1 | | | 0.004–0.04 | | - | | | | | |
| TFIIB Degradation, 20 min | STM | 2 | | 0.04 | 0.004–0.04 | | - | | | | | |
| TFIIB Degradation, 20 min | STM | 3 | | 0.0006 | 0.004–0.04 | | - | | | | | |
| TFIIB Degradation, 20 min | TFO | 1 | - | - | - | - | 0.0008 | | | | | |
| TFIIB Degradation, 20 min | TFO | 2 | - | - | - | - | 0.0008 | 0.0–0.12 | | | | |
| TFIIB Degradation, 60 min | STM | 1 | | | 0.002–0.02 | | - | | | | | |
| TFIIB Degradation, 60 min | STM | 2 | | 0.06 | 0.002–0.02 | | - | | | | | |
| TFIIB Degradation, 60 min | STM | 3 | | 0.0005 | 0.002–0.02 | | - | | | | | |
| TFIIB Degradation, 60 min | TFO | 1 | - | - | - | - | 0.0004 | | | | | |
| TFIIB Degradation, 60 min | TFO | 2 | - | - | - | - | 0.0004 | 0.0–0.24 | | | | |
| TFIIH Inhibition | STM | 1 | | | | - | | | | 0.02 | | |
| TFIIH Inhibition | STM | 2 | | | | - | | 0.0–0.24 | | 0.02 | | |
| TFIIH Inhibition | STM | 3 | | 0.06 | | - | | | | 0.02 | | |
| TFIIH Inhibition | STM | 4 | 0.0005 | | | - | | | | 0.02 | | |
| TFIIH Inhibition | TFO | 1 | - | - | - | - | | | | 0.02 | | |
| TFIIH Inhibition | TFO | 2 | - | - | - | - | | 0.0–0.24 | | 0.02 | | |
| TFIIH Inhibition | TFO | 3 | - | - | - | - | 0.0005 | | | 0.02 | | |

| Experiment | Gene Class | Model # | $k_{1,UAS}$ 0.002 /s | $k_{-1,UAS}$ 0.003 /s | $k_{2,UAS}$ 0.03–0.1 /s | $k_{-2,UAS}$ 0.0–0.03 /s | $k_{3,Promoter}$ 0.002 /s | $k_{-3}$ 0.0–0.03 /s | $k_4$ 0.0075–0.0275 /s | $k_5$ 0.1 /s | $k_6$ 1 kb/min | $k_7$ 0.0325 /s |
|---|---|---|---|---|---|---|---|---|---|---|---|---|
| gcn4pd | STM | 1 | | | | | | 0.0–0.048 | | | | |
| gcn4pd | STM | 2 | | | 0.009–0.09 | | | 0.0–0.039 | | | | |
| gcn4null | STM | 1 | 0.0006 | | | | | | | | | |

to different steps in the transcription cycle. Code for the simulations is available on GitHub, copy archived at *Brickner, 2024*.

## Model fitting

Several parameters in the model were fixed according to previously published data; $k_1$ and $k_{-1}$ were from *Rosen et al., 2020*; $k_5$ was based on the residency time of TFIIH (*Nguyen et al., 2021*); $k_6$ was based on an average elongation rate of 1000 bp/min (*Larson et al., 2011*; *Zenklusen et al., 2008*) and $k_7$ was based on 56±20 s and 70±41 s termination times (*Larson et al., 2011*; *Zenklusen et al., 2008*). Other parameters in the model were free and were fit to either ChEC-seq2 or ChIP-seq data by performing a grid search.

We evaluated each model in the grid by computing the cosine similarity between the output of the model and the empirical data. That is, we calculated the quantity

$$CosineSimilarity = \frac{\sum_i M_i E_i}{\sqrt{\sum_i M_i^2}\sqrt{\sum_i E_i^2}}$$

where $M_i$ is the average occupancy of the model in the $i$th segment (UAS, promoter, transcript, or 3'UTR) and $E_i$ is the corresponding empirical data from the same segment. The cosine similarity ranges from –1 to 1, with 1 indicating perfect alignment, 0 indicating no correlation, and –1 indicating perfect inverse alignment. This measure allows us to quantitatively assess how well each model's predictions align with the observed data simultaneously across gene regions. Rather than choosing the single model with the best fit, we elected to use an ensemble approach to more thoroughly interpret the data. In this approach, all models with cosine similarity greater than 0.995 were included in the ensemble (for ChEC-seq2). This ensemble approach allows us to explore the full space of models that are consistent with the data and avoid any spurious conclusions that may arise from the investigation of a single parameter set. The recovered ensemble of models was distributed across a manifold in parameter space, establishing required relationships between the unknown parameters (*Figure 6—figure supplement 1*, *Figure 7—figure supplement 1*). For ChIP-seq data, the model could not achieve a cosine similarity greater than 0.85, so instead we report the best fitting models to provide context. Genes with fewer than 50 nascent read counts were removed from the STM and TFO datasets, yielding 643 STM genes and 1143 TFO genes.

Based on the established functions of the proteins involved (TFIIB, Kin28, or Gcn4), we identified the rate that would be most likely influenced by the experimental perturbation and simulated the effects of perturbing that rate. If altering that rate was not sufficient to match the data, the effects of changing additional rates were explored to identify the model that best match the data. Changes to rates that did not match the empirical data are not shown. The final list of parameters used to simulate each experiment are given in *Table 2*.

## Data analysis

1. Gene classifications, coordinates, and regions
   The S288C genome sequence and annotations from build R64-3-1 were used for analysis and visualization (*Engel et al., 2014*). The STM and TFO gene classifications are from *Rossi et al., 2021*. TATA-positions were from *Rhee and Pugh, 2012*. The top 150 expressed genes within each class were defined by Nascent RNA counts (SLAM-seq) from the BY4741 strain grown in SDC and are listed in *Supplementary file 1*. Similarly, expressed genes subsets were defined as genes for which there were ≥50 nascent RNA counts on average across three biological replicates. This resulted in the following number of genes per expressed subsets: STM, 643 genes; TFO, 1143 genes; TATA-containing, 597 genes.
   TSS and transcription end site (TES) locations were defined by an RNA-seq dataset (*Pelechano et al., 2013*), when available. In cases where no TSS was available from RNA-seq, the TSS was instead taken from a CAGE-seq dataset (*Lu and Lin, 2021*). If neither dataset contained TSS or TES information, the median 5'UTR length (47 bp) or 3'UTR length (118 bp) was used to define these locations, respectively. Median UTR lengths were calculated from the most abundant transcript isoform for mRNAs (*Pelechano et al., 2013*). ChEC-seq2 signal was binned into gene regions defined as: UAS, –500 bp to –151 relative to TSS; promoter, –150 to +25 relative

to TSS; transcript, +26 relative to TSS and –76 relative to TES; terminator, –75 to +150 relative to TES.

2. Individual gene plots
   A region spanning 1000 bp upstream of the TSS and 1000 bp downstream of the TES is shown. Signal was smoothed with a sliding window average (window = 10, step = 5).

3. Metasite plots
   Genes were aligned by TSS or TATA sequence, as indicated in the figure. 250 bp upstream and downstream of the of the aligned site was included. Signal was smoothed with a sliding window average (window = 10, step = 5).

4. Metagene plots

   Metagene plots are composed of three regions: 1000 bp upstream of the TSS, the transcript (TSS to TES), and 1000 bp downstream of the TES. First, the average signal (or change in signal, where indicated) at each base pair from three biological replicates was calculated. Then, each region was divided into 100 bins and the average signal in each bin was calculated. The process was repeated for each gene, and then the average signal for each bin across all genes was calculated and is displayed in metagene plots.

## Materials availability

The plasmids and strains described in *Supplementary files 2 and 3* are available upon request.

## Acknowledgements

The authors thank Professors Vu Nguyen (University of California, San Diego), David Shore (University of Geneva), Yuan He (NU), Shelby Blythe (NU), Curt Horvath (NU), and Richard Morimoto (NU) for helpful feedback and support, members of the Brickner laboratory for helpful comments on the manuscript and Gabe Zentner for yeast strains, plasmids, and technical advice. DJV was supported by a National Science Foundation Graduate Fellowship and by T32 NIGMS GM008061. BS was supported by National Science Foundation research training grant DMS-1547394. This work was supported by National Institute of General Medical Sciences grant R35GM136419 (JHB).

## Additional information

### Funding

| Funder | Grant reference number | Author |
| --- | --- | --- |
| National Institute of General Medical Sciences | R35GM136419 | Jason H Brickner |
| National Science Foundation | | Jake VanBelzen |
| National Science Foundation | DMS-1547394 | Bennet Sakelaris |
| National Institute of General Medical Sciences | T32GM008061 | Jake VanBelzen |

The funders had no role in study design, data collection and interpretation, or the decision to submit the work for publication.

### Author contributions

Jake VanBelzen, Conceptualization, Data curation, Investigation, Visualization, Writing - original draft, Writing - review and editing; Bennet Sakelaris, Data curation, Software, Formal analysis, Visualization; Donna G Brickner, Nikita Marcou, Investigation; Hermann Riecke, Formal analysis, Supervision, Writing - review and editing; Niall M Mangan, Formal analysis, Supervision; Jason H Brickner, Conceptualization, Data curation, Software, Formal analysis, Supervision, Funding acquisition, Visualization, Writing - original draft, Project administration, Writing - review and editing

## Author ORCIDs

Jake VanBelzen https://orcid.org/0000-0003-0949-3641
Bennet Sakelaris https://orcid.org/0000-0001-8798-584X
Nikita Marcou https://orcid.org/0000-0003-0232-081X
Hermann Riecke https://orcid.org/0000-0002-6070-4742
Niall M Mangan https://orcid.org/0000-0002-3491-8341
Jason H Brickner https://orcid.org/0000-0001-8019-3743

Reviewer #1 (Public review): https://doi.org/10.7554/eLife.100764.3.sa1
Reviewer #2 (Public review): https://doi.org/10.7554/eLife.100764.3.sa2
Author response https://doi.org/10.7554/eLife.100764.3.sa3

## Additional files

### Supplementary files

• Supplementary file 1. Lists of gene subsets used in this study.

• Supplementary file 2. Yeast strains used in this study.

• Supplementary file 3. Plasmids and oligonucleotides used in this study.

• MDAR checklist

### Data availability

Sequencing data has been deposited in the Gene Expression Omnibus at the National Center for Biotechnology Information and can be retrieved with accession numbers GSE267843 and GSE246951. Scripts used in modeling are available at GitHub, copy archived at *Brickner, 2024*.

The following dataset was generated:

| Author(s) | Year | Dataset title | Dataset URL | Database and Identifier |
|---|---|---|---|---|
| Brickner J, VanBelzen DJ | 2024 | ChEC-seq2 of RNA Polymerase II and preinitiation Complex in *S. cerevisiae* | https://www.ncbi.nlm.nih.gov/geo/query/acc.cgi?acc=GSE267843 | NCBI Gene Expression Omnibus, GSE267843 |

The following previously published dataset was used:

| Author(s) | Year | Dataset title | Dataset URL | Database and Identifier |
|---|---|---|---|---|
| VanBelzen DJ, Duan C, Brickner DG, Brickner JH | 2023 | ChEC-seq2: an improved chromatin endogenous cleavage sequencing method and bioinformatic analysis pipeline for mapping in vivo protein-DNA interactions | https://www.ncbi.nlm.nih.gov/geo/query/acc.cgi?acc=GSE246951 | NCBI Gene Expression Omnibus, GSE246951 |

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
