## [Editor Report · eLife Assessment]

This **valuable** study compares ChIP-seq and ChEC-seq2 techniques to investigate RNA polymerase II (RNAPII) binding patterns in yeast, revealing that ChEC-seq2 captures distinct regulatory events associated with active transcription missed by ChIP-seq. The authors use ChEC-seq2 data to build a stochastic model of RNAPII kinetics, providing **convincing** new insights into transcription regulation and the role of the nuclear pore complex. The paper highlights the importance of careful methodological comparisons in understanding RNAPII dynamics.

---

## [Referee Report · Reviewer #1 (Public review)]

Summary:

In this study, the authors use ChEC-seq, an MNase based method to map yeast RNA pol II. Part of the reasoning for this study is that earlier biochemical work suggested pol II initiation and termination should involve slow steps at the UAS/promoter and termination regions that are not well visualized by formaldehyde-based ChIP methods. Here the authors find that pol II ChIP and ChEC give complementary patterns. Pol II ChIP signals are strongest in the coding region (where ChIP signal correlates well with transcription (rho = 0.62)). In contrast, pol II ChEC signals are strongest at promoters (rho = 0.52) and terminator regions. Weaker upstream ChEC signals are also observed at the STM class genes where biochemical studies have suggested a form of Pol (and maybe other general factors) is recruited to UAS sites. ChEC of TFIIA and TFIIE give promoter-specific ChEC signals as expected. Extending this work to elongation factors Ctk1 and Spt5 unexpectedly give strong signals near the PIC location and little signals over the coding region. This, and mapping CTD S2 and S5 phosphorylation by ChEC suggests to me that, for some reason, ChEC isn't optimal for detecting components of the elongation complex over coding regions.

Examples are also presented where perturbations of transcription can be measured by ChEC. Modeling studies are shown where adjustment of kinetic parameters agree well with ChEC data and that these models can be used to estimate which steps in transcription are affected by various perturbations. However, no tests were performed to see if the predictions could be validated by other means. Finally, the role of nuclear pore binding by Gcn4 is explored, although the effects are small and this proposal should be explored more completely in future studies. Overall, the authors show that pol II ChEC is a valuable and complementary method for investigating transcription mechanisms and slow steps at the initiation and termination regions.

---

## [Referee Report · Reviewer #2 (Public review)]

Summary:

The study by VanBalzen et. al. compares chromatin immunoprecipitation (ChIP-seq) and chromatin endogenous cleavage sequencing (ChEC-seq2) to examine RNA polymerase II (RNAPII) binding patterns in yeast. While ChIP-seq shows RNAPII enrichment mainly over transcribed regions, ChEC-seq2 highlights RNAPII binding at promoters and upstream activating sequences (UASs), suggesting it captures distinct RNAPII populations that the authors speculate are linked more tightly to active transcription. The authors develop a stochastic model for RNAPII kinetics using ChEC-seq2 data, revealing insights into transcription regulation and the role of the nuclear pore complex in stabilizing promoter-associated RNAPII. The study suggests that ChEC-seq2 identifies regulatory events that ChIP-seq may overlook.

Strengths:

(1) This is a carefully crafted study that adds significantly to existing literature in this area. Transgenic MNase fusions with endogenous Rpb1 and Rpb3 subunits were carefully performed, and complemented by fusions with several additional proteins that help the authors to dissect the transcription cycle. Both the *S. cerevisiae* lines and the sequencing data are likely to be of significant use to the community

(2) The validation of ChEC-seq2 and its comparison with ChIP-seq is highly valuable technical information for the community.

(3) The kinetic modeling appears to be thoughtfully done.

---

## [Author Response]

The following is the authors’ response to the original reviews.

Summary of revisions

Title

We have changed the title of the manuscript to “Chromatin endogenous cleavage provides a global view of yeast RNA polymerase II transcription kinetics”.

Text

Additional discussion of the patterns for elongation factors added (detailed below).

Small text changes throughout, as mentioned in the detailed response below.

Figures

Updated legend-image in Figure 2F to reflect correct colors

Added Figure 2 – supplement 1F – RNAPII enrichment with shorter promoter dwell times

Added Figure 2 - supplement 2 with ChIP-seq outcomes (and text legend)

Removed gene numbers in Figure 5C and put them in the legend.

Substituted Med1 and Med8 ChEC over Rap1 sites in Figure 5F.

Moved *kin28-is* growth inhibition to Figure 5 – Supplement 1.

Substituted a new panel overlaying the RNAPII enrichment over UASs or promoters for all three strains in Figure 7D.

Improved the labeling and legend of Figure 7E

Methods

Added ChIP-seq performed to confirm that the MNase fusion proteins are able to produce the expected pattern for ChIP.

Point-by-point response to reviewers’ comments

**Reviewer 1:**
(1) Extending this work to elongation factors Ctk1 and Spt5 unexpectedly give strong signals near the PIC location and little signals over the coding region. This, and mapping CTD S2 and S5 phosphorylation by ChEC suggests to me that, for some reason, ChEC isn't optimal for detecting components of the elongation complex over coding regions.(3) mapping the elongation factors Spt5 and Ctk1 by ChEC gives unexpected results as the signals over the coding sequences appear weak but unexpectedly strong at promoters and terminators. It would be helpful if the authors could comment on reasons why ChEC may not work well with elongation factors. For example, could this be something to do with the speed of Pol elongation and/or the chromatin structure of coding sequences such that coding sequence DNA is less accessible to MNase cleavage?(7) The mintbodys are an interesting attempt to measure Pol II CTD modifications during elongation but give unexpected results as the signals in the coding region are lower than at promoters and terminators. It seems like ChIP is still a much better option for elongation factors unless I'm missing something.

We agree with the reviewer that this is a point that could confuse the reader. Therefore, we have devoted two additional paragraphs to possible interpretations of our data in the Discussion:

ChEC with factors involved in elongation (Ctk1, Spt5, Ser2p-RNAPII), when normalized to total RNAPII, showed greater enrichment over the CDS (Figure 3G), as expected. However, it is surprising that we also observed clear enrichment of these factors at promoters (e.g. Figure 3A, E & F). The association of elongation factors with the promoter seems to be biologically relevant. Changes in transcription correlate with changes in ChEC enrichment for these factors and modifications (Figure 4C). Blocking initiation by inhibiting TFIIH kinase led to a reduction of Ser5p RNAPII and Ser2p RNAPII over both the promoter and the transcribed region (Figure 5G). This suggests either that the true signal of these factors over transcribed regions is less evident by ChEC than by ChIP or that ChEC can reveal interactions of elongation factors at early stages of transcription that are missed by ChIP. The expectations for enrichment of elongation factors and phosphorylated CTD are largely based on ChIP data. Because ChIP fails to capture RNAPII enrichment at UASs and promoters, it is possible that ChIP also fails to capture promoter interaction of factors involved in elongation as well.

Factors important for elongation can also function at the promoter. For example, Ctk1 is required for the dissociation of basal transcription factors from RNAPII at the promoter (Ahn et al., 2009). Transcriptional induction leads to increases in Ctk1 ChEC enrichment both over the promoter and over the 3’ end of the transcribed region (Figure 4C). Dynamics of Spt4/5 association with RNAPII from in vitro imaging (Rosen et al., 2020) indicate that the majority of Spt4/5 binding to RNAPII does not lead to elongation; Spt4/5 frequently dissociates from DNA-bound RNAPII. Association of Spt4/5 with RNAPII may represent a slow, inefficient step in the transition to productive elongation. If so, then ChEC-seq2 may capture transient Spt4/5 interactions that occur prior to productive elongation, producing enrichment of Spt5 at the promoter.

(2) Finally, the role of nuclear pore binding by Gcn4 is explored, although the results do not seem convincing (10) In Figure 7, it's not convincing to me that ChEC is revealing the reason for the transcriptional defect in the Gcn4 PD mutant. The plots in panel D look nearly the same and I don't follow the authors' description of the differences stated in the text. In panel A, replotting the data in some other way might make the transcriptional differences between WT and Gcn4 PD mutants more obvious.

The phenotype of the *gcn4-pd* mutant is a quantitative decrease in transcription and this leads to a quantitative decrease, rather than qualitative loss, of RNA polymerase II over the promoter, without impacting the association of RNA polymerase II over the UAS region. This effect is small but statistically significant (p = 4e5). We have changed the title of this section of the manuscript to “ChEC-seq2 suggests a role for the NPC in stabilizing promoter association of RNAPII”. Also, to make comparison clearer, we have plotted the data together in the revised figure (Figure 7D).

The magnitude of the decrease is not large, but we would highlight that is almost as large as that produced by inhibiting the Kin28 kinase (Figure 5H). Because the promoter-bound RNAPII is poorly captured by ChIP, this effect might be difficult to observe by techniques other than ChEC. Obviously, more mechanistic studies will need to be performed to fully understand this phenotype, but this result supports a role for the interaction with the nuclear pore complex in either enhancing the transfer of RNA polymerase II from the enhancer to the promoter or in preventing its dissociation from the promoter.

I think that the related methods cut&run/cut&tag have been used to map elongating pol II. The authors should summarize what is known from this approach in the introduction and/or discussion.

CUT&RUN has been used to map RNAPII in mammals, but we are not aware of reports in *S. cerevisiae*. Work from the Henikoff Lab in yeast mapped transcription factors and histone modifications (PMIDs 28079019 and 31232687). A report using CUT&RUN in a human cell line reported a promoter-5’ bias of RNAPII that appeared to be dependent on fragment length (PMID 33070289). Regardless, the report highlights a key distinction between yeast and other eukaryotes: paused RNAPII. Indeed, paused RNAPII dominates ChIP-seq tracks in metazoans, and so we are hesitant to speculate between CUT&RUN in other species vs. ChEC-seq2 in *S. cerevisiae*.

Are the Rpb1, Rpb3, TFIIA, and TFIIE cleavage patterns expected based on the known structure of the PIC (Figures 2C, E)?

Rpb1 and 3 show peaks at approximately -17 and +34 with respect to TATA. TFIIA (Toa2) shows peaks at -12 and + 12. And TFIIE (Tfa1) shows a peak around +34 (Figure 2C & E):

As shown in the supplementary movie (based on the cMed-PIC structure; PDB #5OQM; Schilbach et al., 2017), upon binding to TBP/TFIID, TFIIA would be expected to cleave slightly upstream and downstream of the protected TATA (-12 and +12), while TFIIE binds downstream after the +12 site is protected and would be closest to the +34 unprotected site (to the right in the image below). RNAPII, which binds the fully assembled PIC, should be able to access either the upstream site (-12) or the downstream site (+34). Rpb1’s unstructured carboxy terminal domain, to which MNase is fused, would give it maximum flexibility, which likely explains why Rpb1 cleaves both at -12 and +34, with a preference for -12. Rpb3 also cleaves both sites, but without an obvious preference.

**Author response image 2. sa3fig2:** Cleavage at -12, +12 and +34.

**Author response image 3. sa3fig3:** Highlighted sites corresponding to the peaks in TFIIA assembled with TBP.

**Author response image 4. sa3fig4:** The complete PIC, protecting the +12 site, but leaving the +34 site exposed.

(6) Figure 2 S1: Pol II ChIP in the coding region gives a better correlation with transcription vs ChEC in promoters. Also, Pol II ChIP at terminators is almost as good as ChEC at promoters for estimating transcription. This latter point seems at odds with the text. The authors should comment on this and modify the text as needed.

Thank you for this comment. We have clarified the text.

In Figures 4 and 5, it's hard to tell how well changes in transcription correlate with changes in Pol II ChEC signals. It might be helpful to have a scatterplot or some other type of plot so that this relationship can be better evaluated.

While we find corresponding increase/decrease in ChEC-seq2 signal in genes identified as up/downregulated by SLAM-seq, the magnitude in change is not well correlated between the two techniques. This was not surprising, because neither ChIP nor ChEC correlate especially well with SLAM-seq (Figure 2 – supplement 1E).

In Figure 5, it's unclear why Pol association with Rap1 is being measured. Buratowski/Gelles showed that Pol associates with strong acidic activators - presumably through Mediator. Rap1 supposedly does not bind Mediator - so how is Pol associating here? Perhaps it would be better to measure Pol binding at STM genes that show Mediator-UAS binding.

Thank you; this is a good point. We chose Rap1 because we had generated high-confidence binding sites in our strains under these conditions by ChEC-seq2. The results suggest that RNAPII is recruited well to these sites and that this recruitment does not require TFIIB. However, in disagreement with the notion that Mediator does not interact with Rap1, ChEC with Mediator subunits Med1 and Med8 also show peaks at these sites (new Figure 5F; the old Figure 5F is now Figure 5 – Supplement 1). Therefore, either these sites are co-occupied by other transcription factors that mind Mediator, or Mediator is recruited by Rap1. In either case, this correlates with binding of RNAPII.

**Reviewer 2**:(1) The term "nascent transcription" is all too often used interchangeably for NET-seq, PRO-seq, 4sUseq, and other assays that often provide different types of information. The authors should make it clear their use of the term refers to SLAM-seq data.

We have clarified throughout the manuscript that nascent transcription measured by SLAM-seq.

The authors should explicitly state that experiments were performed in *S. cerevisiae* in the Results section.

We have made it clear in the title and the text that these experiments were performed in *S. cerevisiae*.

Lines 216-218 state that "None of the 24 predicted the strong signal over the transcribed region with promoter depletion characteristic of ChIP-seq". I understand the authors' point, but there are parameter combinations that produce a flat profile with slightly less signal over the promoter (e.g., 5 sec dwell times and 3000 bp/ min elongation rate). If flanking windows were included, this profile would look something like ChIP-seq. I'd encourage the authors to be more precise with their language.

Thank you for highlighting this over-statement.

We have now clarified the text and added another supplementary panel as follows:

“While some combinations predicted a relatively flat distribution across the gene with lower levels in the promoter, none of the 24 predicted the strong signal over the transcribed region with promoter depletion characteristic of ChIP-seq. Only very short promoter dwell times (*i.e.,* < 1s), produced the low promoter occupancy seen in ChIP-seq (Figure 2 – supplement 1F).”